# Modelling ice sheet evolution and atmospheric $CO_2$ during the Late Pliocene

Constantijn J. Berends[1], Bas de Boer[2], Aisling M. Dolan[3], Daniel J. Hill[3], Roderik S. W. van de Wal[1]

[1]Institute for Marine and Atmospheric research Utrecht, Utrecht University, The Netherlands
[2]Earth and Climate Cluster, Faculty of Science, Vrije Universiteit Amsterdam, The Netherlands
[3]School of Earth and Environment, University of Leeds, United Kingdom

*Correspondence to*: Constantijn J. Berends (c.j.berends@uu.nl)

**Abstract.** In order to investigate the relation between ice sheets and climate in a warmer-than-present world, recent research has focussed on the Late Pliocene, 3.6 to 2.58 million years ago. It is the most recent period in Earth history when such a warm
climate state existed for a significant duration of time. Marine Isotope Stage (MIS) M2 (~3.3 Myr ago) is a strong positive excursion in benthic oxygen records in the middle of the otherwise warm and relatively stable Late Pliocene. However, the relative contributions to the benthic $\delta^{18}O$ signal from deep-ocean cooling and growing ice sheets are still uncertain. Here, we present results from simulations of the late Pliocene with a hybrid ice-sheet—climate model, showing a reconstruction of ice sheet geometry, sea-level and atmospheric $CO_2$. Initial experiments simulating the last four glacial cycles indicate that this
model yields results which are in good agreement with proxy records in terms of global mean sea level, benthic oxygen isotope abundance, ice core-derived surface temperature and atmospheric $CO_2$ concentration. For the Late Pliocene, our results show an atmospheric $CO_2$ concentration during MIS M2 of 233 – 249 ppmv, and a drop in global mean sea level of 10 to 25 m. Uncertainties are larger during the warmer periods leading up to and following MIS M2. $CO_2$ concentrations during the warm intervals in the Pliocene, with sea-level high stands of 8 – 14 m above present-day, varied between 320 and 400 ppmv, lower
than indicated by some proxy records but in line with earlier model reconstructions.

## 1 Introduction

One of the major long-term challenges posed by anthropogenic climate change is sea-level rise due to the large-scale retreat of the Greenland and Antarctic ice-sheets (e.g. Church et al., 2013). However, projecting the magnitude and especially the rate of such a retreat is limited by our understanding of the interactions between global climate and the cryosphere on centennial
to multi-millennial time-scales, especially in a warmer-than-present climate. In order to gain more insight into the behaviour of the Earth system in such a warmer world, numerous recent studies (Bachem et al., 2017; Bragg et al., 2012; Burke et al., 2018; de Boer et al., 2015, 2017; Dolan et al., 2011, 2015; de Schepper et al., 2014; Dowsett et al., 2016; Dwyer and Chandler, 2009; Haywood et al., 2010, 2011, 2013a, 2013b; Hill, 2015; Lunt et al., 2009, 2010, 2012; McKay et al., 2012; Miller et al., 2012; Naish et al., 2009; Naish and Wilson, 2009; Prescott et al., 2014; Sohl et al., 2009; Swann et al., 2018; Tan et al., 2017)
have focussed on the Late Pliocene, 3.6 to 2.58 million years ago, since it is the most recent period in Earth history with

average global temperatures staying warmer than present-day for a significant length of time. Many of these studies, particularly those carried out as part of the Pliocene Modelling Intercomparison Project (PlioMIP; Haywood et al., 2010, 2011), focus on the Mid-Pliocene Warm Period (MPWP), 3.29 – 2.97 Myr ago. This time slab represents a relatively stable period in Earth's climate history with warmer-than-present global temperatures, lasting longer than any of the Quaternary interglacials.

Since it occurred much more recently than other warm periods, such as the late Eocene, the difference in continental configuration with the present is relatively small. Modelling studies generally show a global mean annual surface temperature that was more than 3 °C warmer than the present day (Bragg et al., 2012; Burke et al., 2018; Haywood et al., 2013a, 2013b; Lunt et al., 2010, 2012). Sea surface temperatures were warmer as well, with a strongly reduced meridional gradient leading to a slight warming in the tropics and a strong warming in the polar regions (Bachem et al., 2017; Dowsett et al., 2009; 2013;

2016; Hill, 2015). Sea-level estimates range between 10 and 30 meters above present-day values (de Boer et al., 2017; Dolan et al., 2011; Dowsett et al., 2016; Dwyer and Chandler, 2009; Miller et al., 2011, 2012), caused by the almost complete deglaciation of Greenland and West Antarctica. Estimates of atmospheric $CO_2$ concentrations during this period vary between 250 and 450 ppmv (Badger et al., 2013; Bartoli et al, 2011; Martínez-Botí et al., 2015; Seki et al., 2010; Stap et al., 2016; Zhang et al., 2013).

However, global climate during both the Pliocene in general and the MPWP in particular showed significant variability. In order to describe a more general "warm Earth" state, the MPWP is therefore treated in PlioMIP as an average of several different warm peaks that may or may not have occurred synchronously around the globe (Dowsett et al., 2016; Haywood et al., 2010, 2011). Both directly before the beginning of, and relatively shortly after the end of the MPWP, proxy records indicate

that the Earth experienced periods that were apparently colder than present-day, though neither as cold nor as long in duration as typical Late Pleistocene glaciations.

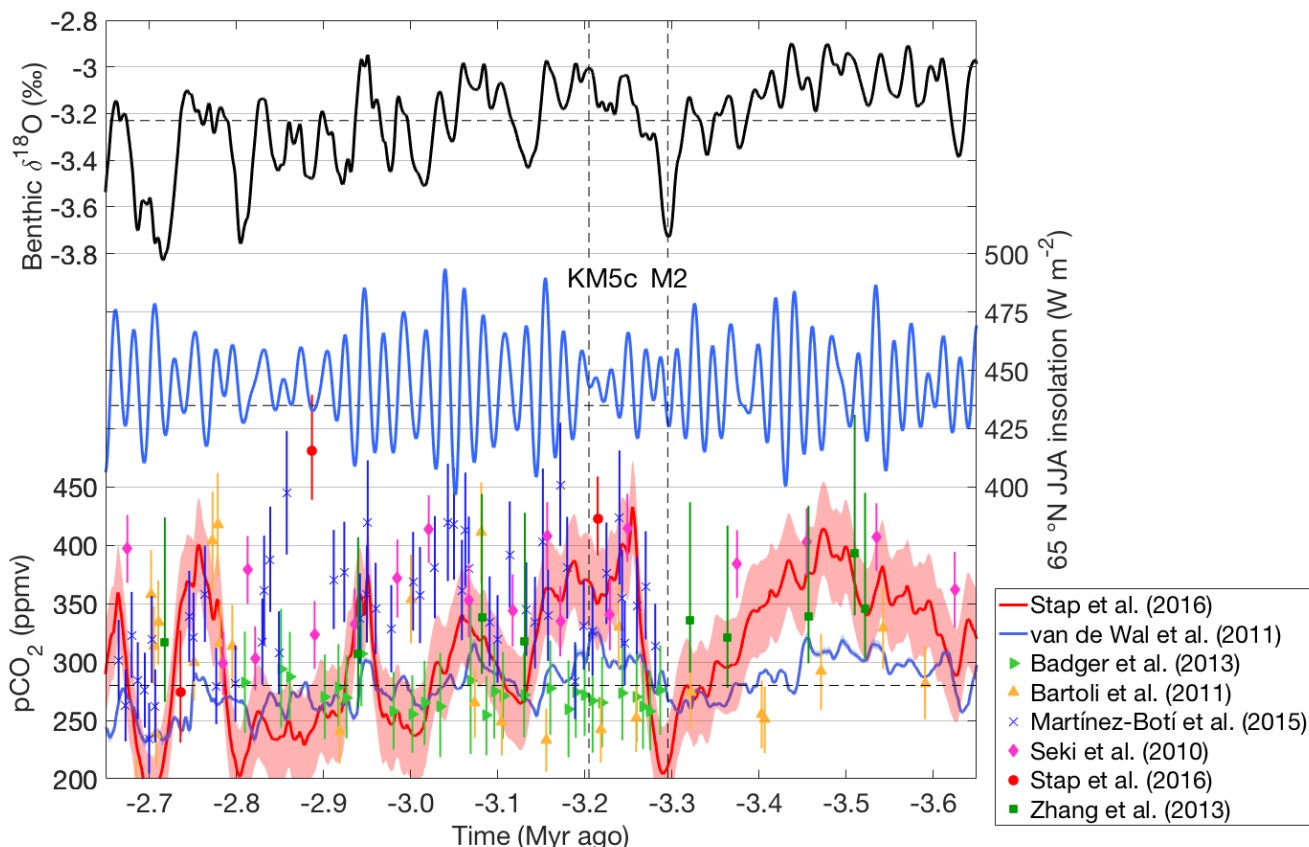

**Figure 1: Benthic δ¹⁸O (LR04; Lisiecki and Raymo, 2005), 65°N summer insolation (Laskar et al., 2004) and reconstructed atmospheric pCO₂ from δ¹⁸O-based model reconstructions (van de Wal et al., 2011; Stap et al., 2016) and proxy data based on alkenones (Seki et al., 2010; Badger et al., 2013; Zhang et al., 2013) and ¹¹B ratios (Seki et al., 2010; Bartoli et al., 2011; Martínez-Botí et al., 2015; Stap et al., 2016) for the late Pliocene. Present-day values for all variables are indicated by horizontal dashed lines, MIS M2 and KM5c are indicated by vertical dashed lines.**

Of particular interest is the cold excursion that occurred 3.3 Myr ago, during Marine Isotope Stage (MIS) M2, shown in Fig. 1. During the 40,000 years following the warm peak of MIS MG1 (3.315 Myr ago), benthic δ¹⁸O (LR04; Lisiecki and Raymo, 2005) increased by about 0.5 ‰ and subsequently recovered, suggesting either a global cooling, an increase in ice volume on the Northern and/or Southern Hemispheres, or both. Sea-level records (Dwyer and Chandler, 2009; 65 ± 25 m; Naish and Wilson, 2009; 38 m; Miller et al., 2011; 34 ± 10 m; Miller et al, 2012; 10 ± 10 m), as well as evidence of glacial till (Gao et al., 2012; de Schepper et al., 2014) and ice-rafted debris (de Schepper et al., 2014; Bachem et al., 2017; Smith et al., 2018) support the hypothesis of at least a partial Northern Hemisphere glaciation. De Schepper et al. (2014) and Dolan et al. (2015) provide detailed overviews of available evidence for glaciation during the Pliocene in general and MIS M2 in particular. However, because most geological fingerprints that would have been left by Pliocene ice-sheets and glaciers would have been overridden or eroded by waxing and waning of the much larger Pleistocene ice-sheets, evidence is limited to mostly the

presence or absence of ice, providing only sparse information on geographical location and little to none on the volumes of these ice sheets.

Dolan et al. (2015) studied MIS M2 from a climatological rather than a glaciological point of view. Using the HadCM3 general circulation model (GCM; Gordon et al., 2000; Valdes et al., 2017; see Sect. 2.1), they performed an ensemble of simulations of global climate during MIS M2 for different postulated and fixed ice-sheet configurations and atmospheric $CO_2$ concentrations. By comparing the results from these different equilibrium simulations to a wide range of available climatological proxies, they attempted to constrain MIS M2 ice volume estimates through the impact such ice-sheets would have on the climate. However, the available proxy records from this era have relatively large uncertainties, and where information is available, it remains difficult to use this to draw sound conclusions about Northern Hemisphere ice sheet extent. They therefore concluded that available evidence from climatological proxies was unable to constrain ice volume any further.

In this study, we adopt a different approach, combining both the glaciological and climatological viewpoints. In a recent study, Berends et al. (2018) presented and evaluated a hybrid GCM – ice-sheet model, where they proposed a matrix method of model coupling (see Sect. 2.3) to force the ANICE ice-sheet model (Bintanja and Van de Wal, 2008; de Boer et al., 2013, 2014, 2017; see Sect. 2.2) with output from the HadCM3 GCM. By using output from a simulation with HadCM3 of the last glacial maximum (Singarayer and Valdes, 2010) they were able to simultaneously simulate the evolution of the ice sheets on North America, Eurasia, Greenland and Antarctica throughout the last glacial cycle and their contributions to global mean sea level and benthic $\delta^{18}O$. They showed that their results matched proxy-based reconstructions for ice-sheet volume, ice surface temperature, sea-water $\delta^{18}O$, deep-water temperature and benthic $\delta^{18}O$. This matrix method is here applied to the late Pliocene by using HadCM3 results from Dolan et al. (2015). The hybrid GCM – ice-sheet model presented by Berends et al. (2018) is computationally efficient enough to make large ensemble simulations feasible, opening up the opportunity to study the effects of changes in paleotopography, $pCO_2$ and other climatological conditions, as well as the sensitivity to ice-sheet model parameters.

However, a high-resolution, time-continuous $pCO_2$ record needed to force the model is not available for this period. We resolve this by using the inverse modelling approach that was also used by Bintanja and van de Wal (2008), de Boer et al. (2013; 2014) and Stap et al. (2016). In this approach we compare modelled benthic $\delta^{18}O$ to the LR04 stack (Lisiecki and Raymo, 2005) and calculate $pCO_2$ based on the difference between the two (see Sect. 2.4). This makes our model set-up conceptually very similar to the approach by Stap et al. (2016), who also used the LR04 stack of $\delta^{18}O$ to force a coupled ice-sheet – climate model and thus produce a $pCO_2$ reconstruction. However, they used a relatively simple zonally averaged energy-balance climate model coupled to a 1-D ice model, whereas we use GCM output to drive 3-D ice-sheet models, making our approach more detailed in terms of the behaviour of global climate, the ice-sheets and the interactions between the two, at the expense of computational

requirements. The $CO_2$ and climate reconstructions by van de Wal et al. (2011), Stap et al. (2016) and the one presented here can be viewed as proxy-based reconstructions, based on the concept that benthic $\delta^{18}O$ is a proxy for changes in ocean temperature and land ice volume. All three studies use a climate model describing the known relations between $pCO_2$ and temperature and ice volume, in order to determine how $pCO_2$ must have evolved in the past in order to produce the observed
benthic $\delta^{18}O$ signal.

## 2 Methodology

### 2.1 Climate model

HadCM3 is a coupled atmosphere-ocean general circulation model (Gordon et al., 2000; Valdes et al., 2017). It accurately reproduces the heat budget of the present-day climate (Gordon et al., 2000) and has been used for future climate projections
in the IPCC AR4 (e.g. Solomon et al., 2007), and paleoclimate reconstructions such as PMIP2 (Braconnot et al., 2007) and PlioMIP (Haywood and Valdes, 2003; Dolan et al., 2011, 2015; Haywood et al., 2013). The atmosphere module of HadCM3 has a resolution of 2.5 ° latitude by 3.75 ° longitude. The ocean is modelled at a horizontal resolution of 1.25 ° by 1.25 °, with 20 vertical layers. In the model set-up by Berends et al. (2018), the climate matrix consists of two GCM snapshots, of respectively the pre-industrial period (PI) and the last glacial maximum (LGM), produced by Singarayer and Valdes (2010)
with HadCM3. Here, we include several additional snapshots focussing specifically on the Pliocene.

### 2.2 Ice-sheet model

To simulate the evolution of the ice sheets we use ANICE, a coupled 3-D ice-sheet-shelf model (Bintanja and Van de Wal, 2008; de Boer et al., 2013, 2014, 2017). It combines the shallow shelf approximation (SSA; Morland, 1987) for floating ice shelves with the shallow ice approximation (SIA; Morland and Johnson, 1980) for grounded ice to solve the ice flow. A Mohr-
Coulomb plastic law for basal sliding is included, with basal stresses included in the SSA equations. The basal stress is calculated as a function of a till stress, which in turn depends on the local bedrock elevation (Winkelmann et al., 2011; de Boer et al., 2013). For grounded ice, the velocities resulting from both approximations are summed, resulting in a smooth transition zone between slow-flowing land ice and fast-flowing floating ice. This approach allows the grounding line to respond to changes in shelf buttressing, resulting in proper glacial-interglacial differences in Antarctic ice volume (de Boer et al., 2013;
Berends et al., 2018) by the advance of grounding lines in the Filcher-Ronne and Ross basins toward the continental shelf. The surface mass balance is parameterised using an insolation-temperature scheme using monthly temperatures and precipitation, refreezing of water and a correction for orographic forcing of precipitation; a more detailed model description is provided by de Boer et al. (2013) and references therein. The horizontal resolution of ANICE for this application is 20 km for Greenland and 40 km for the other three regions (North America, Eurasia and Antarctica). The highly parameterized climate forcing and
resulting computational efficiency of ANICE allow for transient simulations of multiple glacial cycles to be carried out within

10 – 100 hours on single-core systems, making ensemble simulations feasible. Melt underneath the ice shelves is calculated using a linear relation to ocean temperature change (Pollard and DeConto, 2009; Martin et al., 2011), a parameterisation of sub-shelf cavity circulation based on the shortest linear distance to the open ocean (Pollard and DeConto, 2009), and the glacial–interglacial variance parameterization by Pollard and DeConto (2009). A more detailed explanation is provided by de

Boer et al. (2013), who tuned this approach to produce realistic present-day Antarctic shelves and grounding lines. A simple threshold thickness of 200 m is used to describe ice calving, whereby any shelf ice below this thickness is removed.

## 2.3 Matrix method

Using the definition by Pollard (2010), a climate matrix is a collection of pre-calculated output data from several steady-state GCM simulations, called "snapshots", that differ from each other in one or more key parameters, such as prescribed

atmospheric $pCO_2$, orbital configuration or ice-sheet configuration, each creating a separate dimension of the matrix. When performing a simulation with an ice-sheet model, at every point in time during the simulation the prescribed climate forcing is determined by combining the climate states constituting the matrix according to the position of the model state within the matrix. This constitutes a middle ground between methods of offline forcing, such as a glacial index method, and fully coupled ice-sheet – climate models. When the ice-sheet model is in a state corresponding to one of the GCM snapshots, the climate

from this snapshot will be prescribed, containing the effects of the altitude-temperature and albedo-temperature feedbacks of the ice-sheets, the effect of ice sheet geometry on large-scale atmospheric circulation and precipitation, and possibly also the effects of changed freshwater fluxed on ocean circulation, depending on the GCM. When the ice-sheet model is in a state lying in between different GCM snapshots, the prescribed climate is an interpolation of these snapshots. This means that non-linearities in different feedback processes, such as the effects of ice-sheet geometry on atmospheric circulation and

precipitation, are difficult to properly account for.

In this study, we use the model set-up developed by Berends et al. (2018), who created a matrix with the HadCM3 climate states of the pre-industrial and the last glacial maximum from Singarayer and Valdes (2010) and used it to force the ANICE ice-sheet model. In this set-up, temperature fields from the two climate states are combined based on a prescribed value for

$pCO_2$ and on the internally modelled ice-sheets, with the feedback of the ice sheets on the climate based via the effect on absorbed insolation through changes in surface albedo. This interpolation is carried out separately for all four ice sheets. The altitude-temperature feedback is parameterised by a constant lapse-rate derived from the GCM snapshots. Precipitation fields are combined based on changes in surface elevation, reflecting the orographic forcing of precipitation and resulting plateau desert caused by the presence of a large ice-sheet. Berends et al. (2018) demonstrated the viability of this method by simulating

the evolution of the North American, Eurasian, Greenland and Antarctic ice-sheets throughout the entire last glacial cycle, showing that model results agree well with available data in terms of ice-sheet extent, sea-level contribution, ice-sheet surface temperature and contribution to benthic $\delta^{18}O$.

In this study, we extended the PI-LGM climate matrix by adding several climate states from the study by Dolan et al. (2015). The four different ice sheet configurations they used are shown in Fig. 2. The "PRISM" ice sheets are based on the PRISM3 reconstruction (Dowsett et al., 2010), which is a time-slab representation of average peak warm conditions during the MPWP. The "Small" ice sheets are present-day conditions. The "Medium" and "Large" ice sheets were based on the ICE-5G

reconstruction of the last deglaciation (Peltier, 2004) at 8 ky ago and 11 ky ago, respectively. Each of these configurations was used as boundary conditions for two simulations with HadCM3: one with 280 ppmv and one with 220 ppmv $pCO_2$, both with 3.3 My orbital parameters. This adds up to eight different snapshots, plus one additional "Plio_Control" simulation with the PRISM3 ice sheet configuration and orbital parameters, and 405 ppmv $pCO_2$. These simulations allow the climate matrix to separate effects on climate by $pCO_2$ and ice-sheet extent and provide valuable information on climates that are both warmer

and colder than present-day. Although the new climate matrix is relatively sparse for warmer-than-present worlds, containing only one snapshot (Plio_Control) for $pCO_2 > 280$ ppmv, and only three snapshots for smaller-than-present ice sheets (Plio_Control, PRISM_280 and PRISM_220), we believe it is still suitable for simulating the warm Pliocene. The matrix used by Berends et al. (2018) to simulate the last glacial cycle only contained two GCM snapshots in total, and still produced satisfactory results.

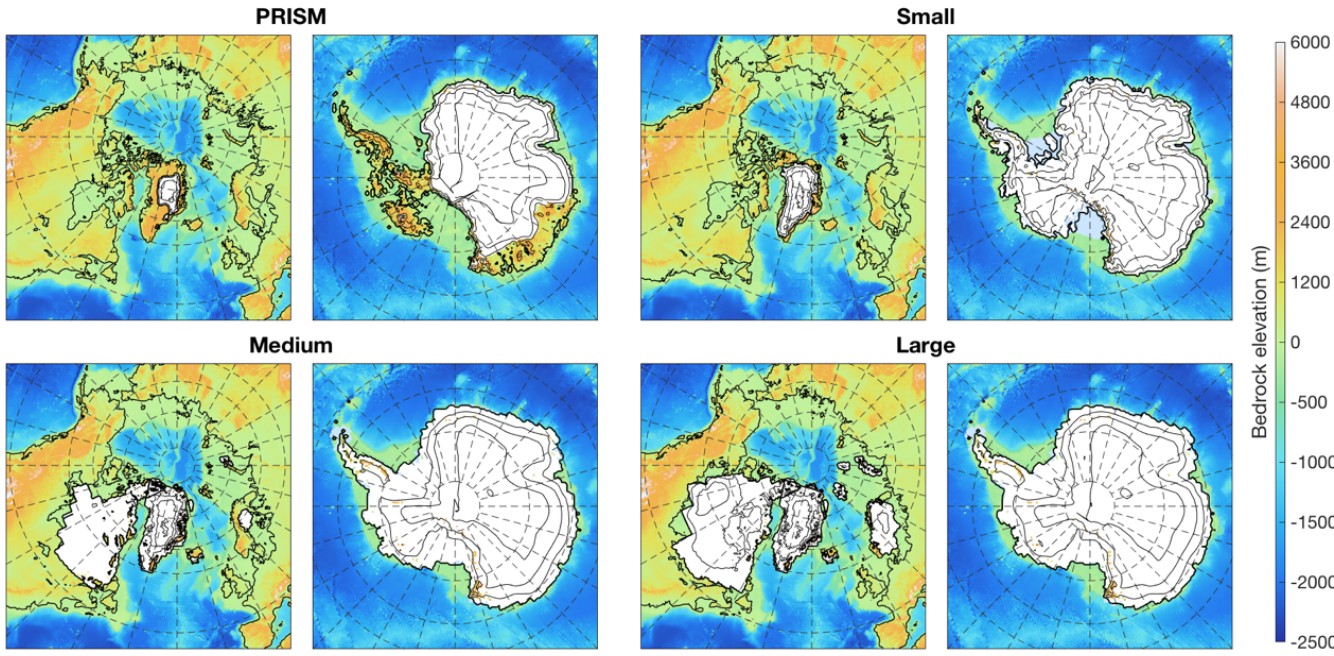

**Figure 2: The four ice-sheet configurations used by Dolan et al. (2015) as boundary conditions for their HadCM3 simulations. "PRISM" is the PRISM3 ice sheet from Dowsett et al. (2010), "Small" is present day, "Medium" is the ICE-5G reconstruction (Peltier, 2004) at 8 ky ago and "Large" is ICE-5G at 11 ky ago.**

This extended matrix therefore allows for a more accurate simulation of both the warm Late Pliocene and the cold MIS M2

glaciation. For the North America and Eurasia modules of ANICE, we added the simulations of the Medium and Large ice-

sheets (both the 280 and 220 ppmv $pCO_2$ versions) by Dolan et al. (2015), since those provide extra information on the effect on climate of intermediate-sized ice-sheets, as well as the Plio_Control simulation for its information on $pCO_2$ levels above 280 ppmv. In the case of North America and Eurasia, we did not use the Small and PRISM states as there is no ice on either continent, meaning these simulations contain no additional constraints for these ice-sheet models. For the Greenland and

Antarctica models, we chose to only add the simulations of the two PRISM states and the Plio_Control simulation, because they provide new information on the effect on climate of smaller-than-present-day ice-sheets. The Medium and Large simulations were left out of the matrix because the ICE-5G ice-sheets (Peltier, 2004) that were used to force those HadCM3 simulations have the exact same horizontal extent as the ICE-5G LGM ice-sheets. Not only does this make it difficult to distinguish between these states in the interpolation routines, it also means the effect on local climate, other than through the

altitude-temperature feedback, is likely to have been small.

## 2.4 Inverse method

The inverse forward modelling approach used to determine $pCO_2$ based on the difference between modelled and observed benthic $\delta^{18}O$ is very similar to that described by de Boer et al. (2013). Their method calculates how the climate at high latitudes, described by a single, spatially uniform temperature offset $\Delta T_{NH}$, should have evolved, such that its effect on deep ocean

temperature and land ice volume reproduces the observed benthic $\delta^{18}O$ signal. This is achieved by comparing the modelled benthic $\delta^{18}O$ value $\delta^{18}O_{mod}$ at every time step to the observed value $\delta^{18}O_{obs}$. If it is too positive, then either the ocean is not cold enough or there is not enough land ice. Global mean surface temperatures are then lowered in the next time step, leading to both a cooling in the deep ocean and an increase in ice growth. This relation is quantified by de Boer et al. (2013) by the following equation:

$$\Delta T_{NH} = \overline{\Delta T_{NH}} + 20 \left( \delta^{18}O_{mod} - \delta^{18}O_{obs}(t + 0.1\ ky) \right). \tag{1}$$

Here, $\overline{\Delta T_{NH}}$ is the mean surface temperature anomaly between 40 and 80 degrees latitude at sea level over the preceding 2 kyr. The modelled benthic $\delta^{18}O$ is calculated using ice volume, ice-sheet $\delta^{18}O$ and deep-water temperatures relative to PD for every 100 years. The optimum values of 2 kyr for the length of the averaging window and 20 for the scaling parameter were

determined by de Boer et al. (2013), producing a value of $\Delta T_{NH}$ = -15 K at LGM.
Since the climate matrix used in our model determines the regional climate based on the modelled ice sheet and the scalar atmospheric CO2 concentration, we adapted this approach for our model set-up by using the difference between modelled and observed $\delta^{18}O$ to calculate a value for $pCO_2$, which is subsequently forwarded to the climate matrix. That way, our model reconstructs how $pCO_2$ should have evolved in order to change global climate in such a way that the resulting changes in deep

ocean temperature and land ice volume reproduce the observed benthic $\delta^{18}O$ signal. The algorithm then becomes:

$$pCO_2 = \overline{pCO_2} + 120\left(\delta^{18}O_{mod} - \delta^{18}O_{obs}(t + 0.1\ ky)\right). \tag{2}$$

As the constrained quantity is the change in $pCO_2$, the scaling factor changes to 120 ppmv / per mill $\delta^{18}O$ change in order to produce a glacial interglacial contrast of 90 ppmv $pCO_2$. Based on the results of preliminary experiments, the length of the $CO_2$ averaging time window was increased to 8.5 kyr, in line with the higher values given by Stap et al. (2016). For forcing the inverse routine, the LR04 stack was used. Although a few different globally distributed stacks are available (e.g. Imbrie et al., 1984; Lisiecki and Raymo, 2005; Zachos et al., 2001, 2008; Cramer et al., 2009), the differences among them are small for the late Pliocene and the Pleistocene. In order to maintain consistency with earlier reconstructions based on inverse modelling methods (de Boer et al., 2013; Stap et al., 2016; van de Wal et al., 2011) we decided to use the LR04 stack.

A conceptual visualisation of the inverse-method forced matrix model is shown in Fig. 3.

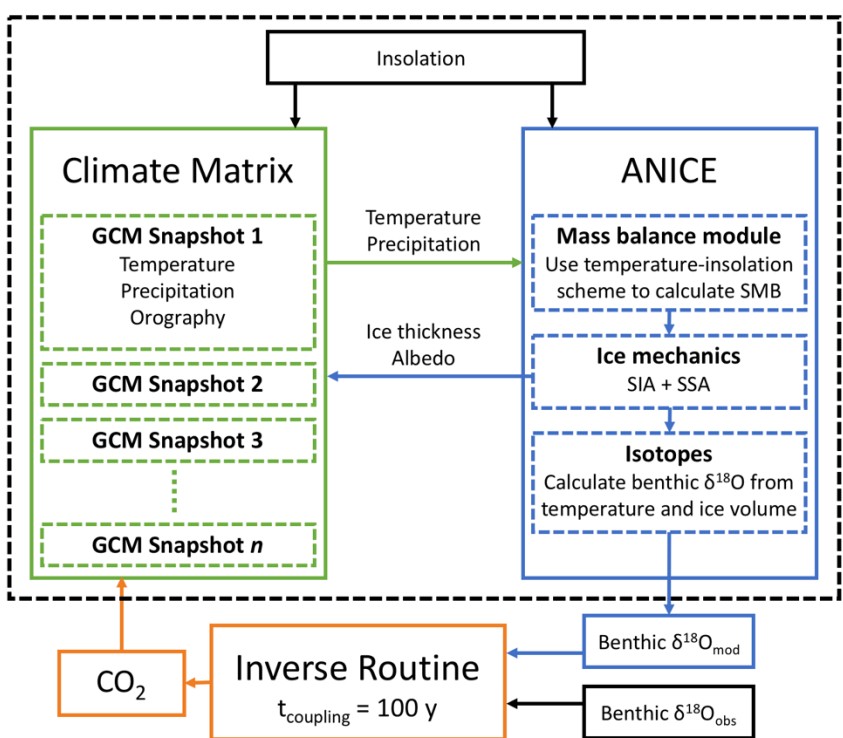

**Figure 3: A conceptual visualisation of the inverse forward modelling approach. The model is forced externally by a benthic $\delta^{18}O$ record and an insolation reconstruction (black boxes). The inverse routine calculates $pCO_2$ based on the difference between observed and modelled $\delta^{18}O$. This value is forwarded to the climate matrix, which interpolates between the GCM snapshots based on the prescribed $pCO_2$ value and the modelled state of the cryosphere (ice thickness and albedo).**

**2.5 Paleotopography reconstruction**

Several recent studies have investigated the evolution of bedrock topography in the geological past. Although ANICE does include a regional solid Earth model to calculate vertical bedrock movement in response to changes in ice distribution, other processes such as erosion and plate tectonics are currently not accounted for within the model, and require external forcing.

For the last glacial cycle benchmark simulation, such effects are assumed to be negligibly small, but this assumption might no longer be valid when going millions of years back in time.

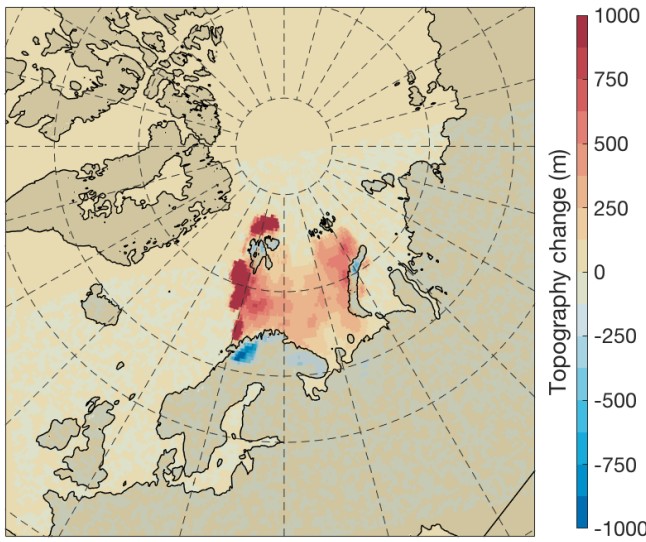

**Figure 4: Topography change relative to present-day for the reconstruction created by Butt et al. (2002) and used by Hill (2015).**

In our study, we use the paleotopography reconstruction of the Barents Sea area by Butt et al. (2002), shown in Fig. 4, based

on a reversal of the erosion of sediments by the Pleistocene ice-sheets. In order to investigate the effect on atmospheric and oceanic circulation, Hill (2015) used HadCM3 to perform simulations both with this topography reconstruction and with present-day topography, both at 405 and 220 ppmv $pCO_2$. By subtracting the calculated climate fields (temperature and precipitation) for the paleotopography simulation from the present-day topography simulation, and adding the resulting "fingerprint" to the climate fields generated by our climate matrix, we take into account the effect of this change in topography

(and the accompanying change in the land/ocean mask) on the global climate.

Preliminary experiments showed that forcing the ice-sheet model with the paleotopography reconstruction without applying this climate "fingerprint" resulted in the persistent presence of a small (~5 m sea-level equivalent) ice-sheet over the newly exposed Barents Land. The climate fingerprint obtained from the simulations by Hill (2015) changes the local climate from an oceanic to a continental climate, with colder, dryer winters and warmer, wetter summers, resulting in more summer melt and

an overall more negative mass balance, implying less ice.

Although other areas of the world might have been eroded by ice sheets (i.e. the Canadian Archipelago, Dowsett et al., 2016; Antarctica, Wilson and Luyendyk, 2009), no GCM simulations investigating the effect on global climate of reversing those changes are currently available. We have therefore chosen not to apply any of these other topography reconstructions to our model.

## 3 Results

### 3.1 Last glacial cycle benchmark

In order to assess the performance of the model when calculating $pCO_2$ with the inverse routine instead of prescribing it directly from an ice core record, we first performed a simulation of the last four glacial cycles, similar to the work by Berends et al. (2018). The model was calibrated by tuning the ablation parameter for the four individual ice-sheets such that their volumes at LGM match the ICE-5G reconstruction. We then performed a sensitivity analysis similar to the experiment described by Berends et al. (2018), investigating the sensitivity of the modelled sea-level drop and benthic $\delta^{18}O$ to the uncertainty in the prescribed forcing (range based on the uncertainty reported by Lisiecki and Raymo, 2005), the ablation tuning parameter (range based on the allowed values found by Berends et al., 2018) and SIA/SSA enhancement factors (increasing the ice velocities calculated for isotropic ice to more closely match those calculated for anisotropic ice according to the approach by Ma et al. (2010); ranges based on the allowed values reported by Ma et al., 2010), as well as several new model parameters involved in the inverse forcing method: the averaging time for the modelled $pCO_2$ (range based on the values reported by de Boer et al. (2013) and Stap et al. (2016)), the ratio between surface temperature anomaly and deep-sea water temperature anomaly and the tuning parameter relating $pCO_2$ to the difference between observed and modelled $\delta^{18}O$ (ranges based on the values reported by de Boer et al. (2010) and Bintanja and van de Wal (2008)), resulting in 17 individual simulations. The values that were used for all these parameters are listed in Table 1. The 17 ensemble members thus yield an estimate of the uncertainty related to both model parameters and forcing.

The simulated $pCO_2$ record is compared to the EPICA Dome C ice core record (Lüthi et al., 2008) in Fig. 5. The ranges of modelled values for $pCO_2$ and sea-level drop at LGM for all investigated model parameters are listed in Table 2. Based on these uncertainties, the model shows that LGM pCO2 is 188 – 197 ppmv and that the sea-level equivalent volume of the four continental ice-sheets at LGM was 83 – 100 m, agreeing well with the values of 185 ppmv $pCO_2$ from the EPICA ice core and 100 m sea-level equivalent ice volume from the ICE-5G reconstruction (Peltier, 2004). The modelled $pCO_2$ values match the EPICA record better than the values simulated by Stap et al. (2016), as demonstrated by the linear correlation and root mean squared error (RMSE) between the EPICA Dome C record and the reconstructions; $R^2 = 0.46$ and RMSE = 23.7 ppmv for Stap et al. (2016) and $R^2 = 0.71$ and RMSE = 15.2 ppmv for our simulation. The reconstruction by van de Wal et al. (2011) performs very similarly to ours ($R^2 = 0.72$, RMSE = 14.7 ppmv), but since it was partly derived from the EPICA record, the comparison is not independent and therefore can not be compared to our results.

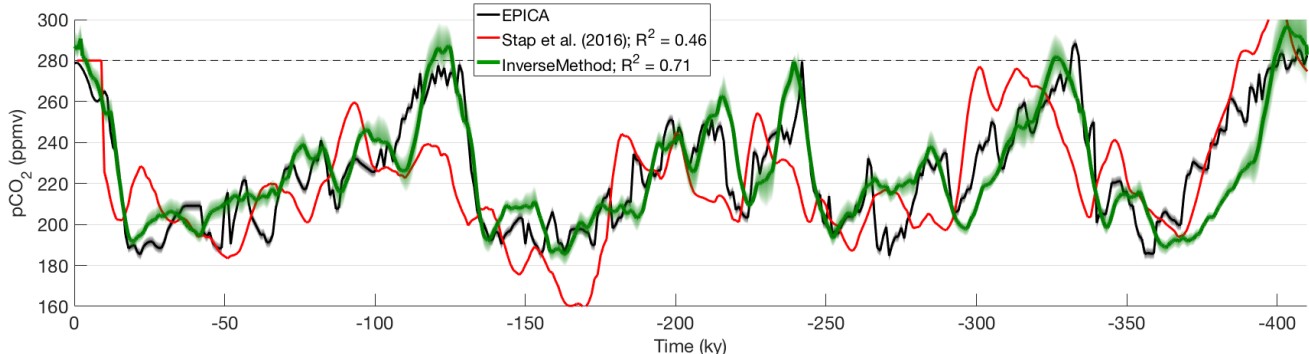

**Figure 5: pCO₂ throughout the last four glacial cycles (410 ky ago - PD): observations from the EPICA Dome C ice core (Lüthi et al., 2008), reconstruction by Stap et al. (2016) and results from the inverse-method forced matrix model (this study). Solid green line shows the benchmark run, green shaded area shows the maximum uncertainty range from the sensitivity experiment with 17 ensemble members, the dotted line indicates the pre-industrial CO₂ concentration. Linear correlation coefficients R² are shown for the correlation between modelled pCO₂ and the EPICA Dome C record.**

Benthic oxygen isotope abundance and its contributions from ice volume and deep-sea water temperature are shown in Fig. 6 and compared to reconstructions by Lisiecki and Raymo (2005) and by Shakun et al. (2015), who made a proxy-based decomposition of the respective contributions to the benthic $\delta^{18}O$ from land ice and deep-sea temperature. The simulated benthic $\delta^{18}O$ shows a near perfect match with the LR04 stack (Lisiecki and Raymo, 2005) that was used to force the model, as is to be expected when using the inverse forward modelling approach. The observed rapid drop in $\delta^{18}O_{sw}$ at the inception, between 120 and 110 ky BP, is reproduced well, as is the drop in $T_{dw}$. Deep water temperature between 60 ky and LGM appears to be too high, more so than for the CO₂-forced model version by Berends et al. (2018).

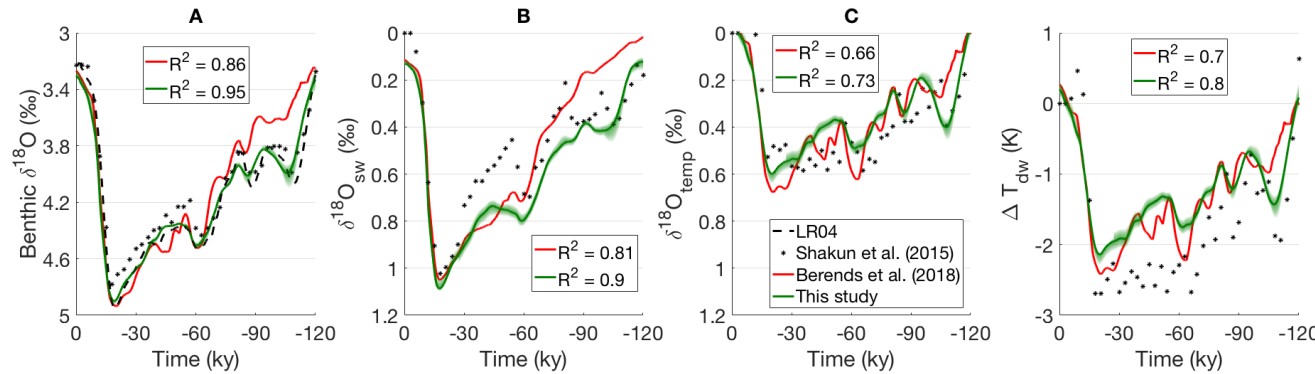

**Figure 6: Benthic $\delta^{18}O$ for the LGC simulation using the inverse-method forced matrix model, compared to data from LR04 (A; Lisiecki and Raymo, 2005) and from Shakun et al. (2015) for both the contribution from the $\delta^{18}O$ of seawater (B) and that from deep water temperature (C), as well as the resulting deep water temperature itself (D). Also shown are the results from Berends et al. (2018). Solid green line shows the benchmark run, green shaded area shows the maximum uncertainty range from the sensitivity experiment. Since the new model set-up is forced with the LR04 stack, rather than with the EPICA CO₂ record used by Berends et al. (2018), the increased correlation coefficients are not a strong result.**

Surface temperature anomalies over Greenland and Antarctica compared to ice-core records (EPICA Dome C; Jouzel et al., 2007; GISP2; Alley, 2000; NGRIP; Kindler et al., 2014) throughout the last glacial cycle are shown in Fig. 7. The performance of the new model version in terms of ice surface temperature is comparable to that of the model by Berends et al. (2018), as illustrated by the linear correlation coefficients and root mean square error between the modelled temperatures and the ice core

records: $R^2 = 0.87$ and RMSE = 0.86 K for Antarctica in this model, versus $R^2 = 0.84$ and RMSE = 0.91 K in the old version. For Greenland, the new model produces a value of $R^2 = 0.74$ and RMSE = 2.2 K, versus $R^2 = 0.65$ and RMSE = 2.6 K for the old model.

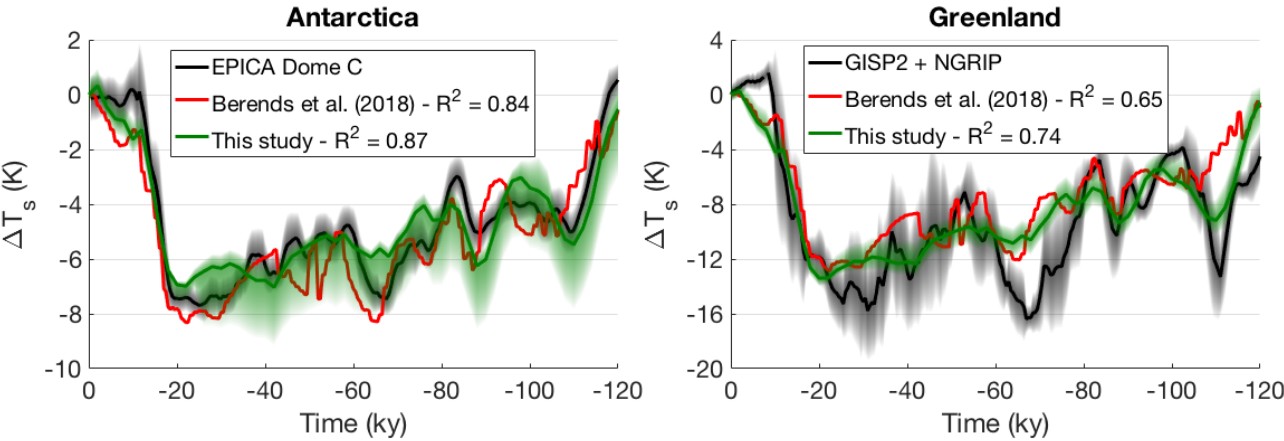

**Figure 7: Modelled versus reconstructed surface temperature anomaly ΔT$_s$ for Antarctica (EPICA Dome C; Jouzel et al., 2007) and**
**Greenland (GISP2; Alley, 2000; NGRIP; Kindler et al., 2014) for the LGC simulation using the inverse-method forced matrix model, compared to the direct pCO$_2$-forced matrix model by Berends et al. (2018). Solid green line shows the benchmark run, green shaded area shows the maximum uncertainty range from the sensitivity experiment. Ice-core temperature records have been subjected to a 4 ky running average; variance shown by black shaded area. Linear correlation coefficients $R^2$ are shown for the correlation between modelled ice surface temperatures and ice core records.**

The mismatch during the inception of the glacial cycle between isotope-derived Antarctic surface temperature and ice core $CO_2$ on the one hand and benthic $\delta^{18}O$ and sea level on the other hand, reported by Bintanja and van de Wal (2008), van de Wal et al. (2011), de Boer et al. (2014), Niu et al. (2017) and Berends et al. (2018), is much better in the simulations here. The linear correlation coefficient $R^2$ between modelled and reconstructed Antarctic surface temperatures between 120 and 80 ky ago increased from a value of 0.49 for Berends et al. (2018) to a value of 0.74 for this study. For Greenland, this value increased

from 0 to 0.36. The $CO_2$-forced model (Berends et al., 2018) produced Antarctic surface temperatures that were in good agreement with the isotope-based proxy record but failed to reproduce the strong sea-level drop. The $\delta^{18}O$-forced model from this study reproduces benthic $\delta^{18}O$ and its different contributions and shows a too strong decrease in $pCO_2$, but is in overall agreement with proxy records of $CO_2$, sea level and temperature, indicating that there is an added value of using the climate matrix method as applied here.

## 3.2 Transient simulation of the Pliocene

The comparisons between model results and (proxy) data for the simulations of the last four glacial cycles indicate that the model accurately reproduces $pCO_2$, ice volume and general geometry (not shown), and surface temperatures. We therefore proceeded to apply the new model set-up to the late Pliocene. We chose to start our transient simulations at 3.65 My ago, capturing the warm period between 3.6 and 3.4 My. The simulations were run until 2.75 My ago, since the density of available $pCO_2$ proxy data is much higher after MIS M2, allowing for a more detailed comparison of modelled $pCO_2$ to proxy-based $pCO_2$ reconstructions. The model was initialised with the same PRISM3 ice-sheets (Dowsett et al., 2010) that were also used to force the PRISM and Plio_Control HadCM3 experiments by Dolan et al. (2015). Due to the nature of the inverse coupling method, initialising the model with present-day ice-sheets quickly converges to the same result. Topography was set to present-day plus the Barents Sea erosion reversal from Butt et al. (2002) and a glacial isostatic adjustment (GIA) correction accounting for the difference in ice loading over Greenland and Antarctica according to the PRISM3 reconstruction (Dowsett et al., 2010). Insolation and benthic $\delta^{18}O$ were prescribed according to Laskar et al. (2004) and Lisiecki and Raymo (2005), respectively. In order to estimate the uncertainty in the modelled ice volume, we performed the same sensitivity analysis as for the last glacial cycle, with the same parameter values shown in Table 1. The resulting simulated $pCO_2$ record is shown in Fig. 8 and compared to other model reconstructions (van de Wal et al., 2011; Stap et al., 2016) and to proxy-based data derived from alkenones (Seki et al., 2010; Badger et al., 2013; Zhang et al., 2013) and $\delta^{11}B$ ratios (Seki et al., 2010; Bartoli et al., 2011; Martínez-Botí et al., 2015; Stap et al., 2016).

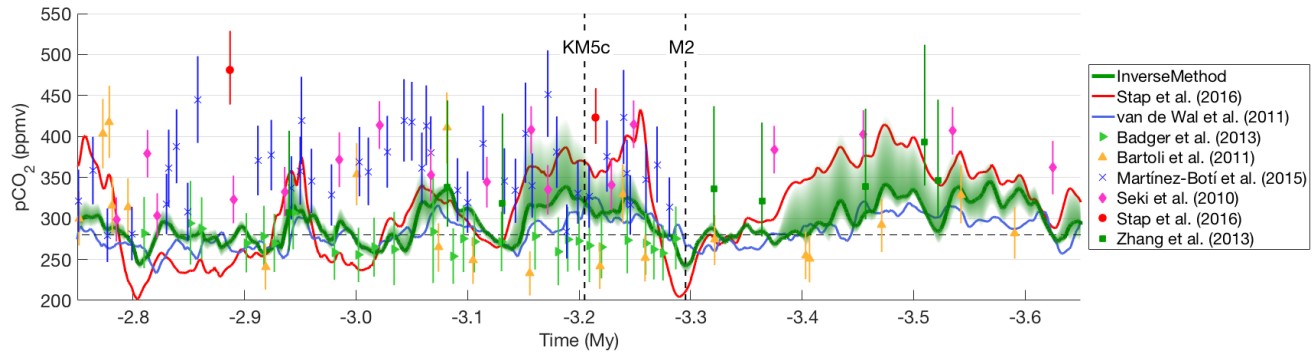

**Figure 8: $pCO_2$ throughout the late Pliocene and early Pleistocene as simulated with the inverse-method forced matrix model, compared to $\delta^{18}O$-based model reconstructions (van de Wal et al., 2011; Stap et al., 2016) and proxy data based on alkenones (Seki et al., 2010; Badger et al., 2013; Zhang et al., 2013) and $^{11}B$ ratios (Seki et al., 2010; Bartoli et al., 2011; Martínez-Botí et al., 2015; Stap et al., 2016). Solid line shows the benchmark run, shaded area shows the maximum uncertainty range from the sensitivity experiment.**

The ranges of modelled values $pCO_2$ and sea-level change at MIS M2 and at KM5c (3.205 My ago) for all investigated model parameters are listed in Table 2. KM5c is used because it has been identified as a time slice representing the mid-Pliocene Warm Period (Haywood et al., 2013). These simulations show that during MIS M2 $pCO_2$ is 233 – 249 ppmv and that the sea-level equivalent volume of the four continental ice-sheets was 10 – 25 m bigger than present-day, with the uncertainty based

on the spread in the results from the ensemble of simulations. The uncertainty in modelled $pCO_2$ becomes much larger for warmer-than-present climates, as shown by the modelled ranges for KM5c. The sensitivity to the benthic $\delta^{18}O$ forcing is especially high, resulting in modelled pCO2 values of 303 – 384 ppmv. The reason for this is that the climatological forcing resulting from the climate matrix is less constrained for warmer than present-day climates. Whereas the matrix contains six snapshots describing climates with more ice and/or lower pCO2, there is only one ice sheet configuration smaller than present-day (PRISM), and only one snapshot with a pCO2 higher than 280 ppmv (the Plio_Control simulation, with 405 ppmv).

The resulting modelled sea-level contributions over time are shown in Fig. 9. The modelled ice-sheets over the Northern and Southern Hemispheres at MIS M2 and KM5c are shown in Fig. 10 and Fig. 11, respectively. In North America, MIS M2 is clearly visible as a strong peak in ice volume, which immediately disappears when pCO2 rises again. Most of the ice forms over north-eastern Canada, with a smaller ice sheet developing over the northern Cordillera. In Eurasia, only small ice-caps form on Svalbard and Nova Zembla (no longer islands, but now small mountain areas bordering the newly exposed Barents Land), with no sizeable ice sheets forming even at the peak of MIS M2. Greenland is mostly ice-free until MIS M2, when it rapidly develops an ice sheet slightly larger than present day. After MIS M2, the ice sheet disappears, advancing and retreating several times during the following period. Similar behaviour is observed on West Antarctica, while East Antarctica remains stable throughout the simulation. While both Greenland and Antarctica continue to show substantial variability throughout the remainder of the simulations, North America does not glaciate again until the onset of the Pleistocene glacial cycles, 2.8 My ago.

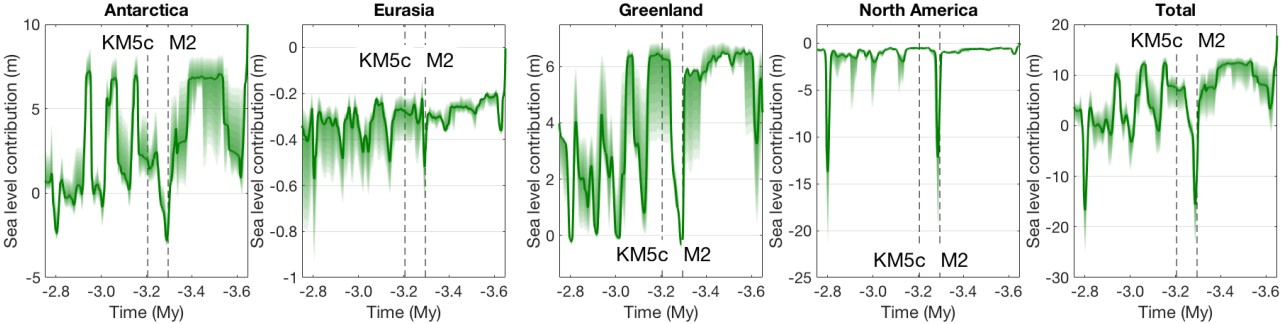

Figure 9: Volumes of the four ice sheets over time throughout the late Pliocene and early Pleistocene as simulated with the inverse-method forced matrix model. Solid green line shows the benchmark run, green shaded area shows the maximum uncertainty range from the sensitivity experiment. Vertical dashed lines indicate MIS M2 (3.295 My ago) and KM5c (3.205 My ago).

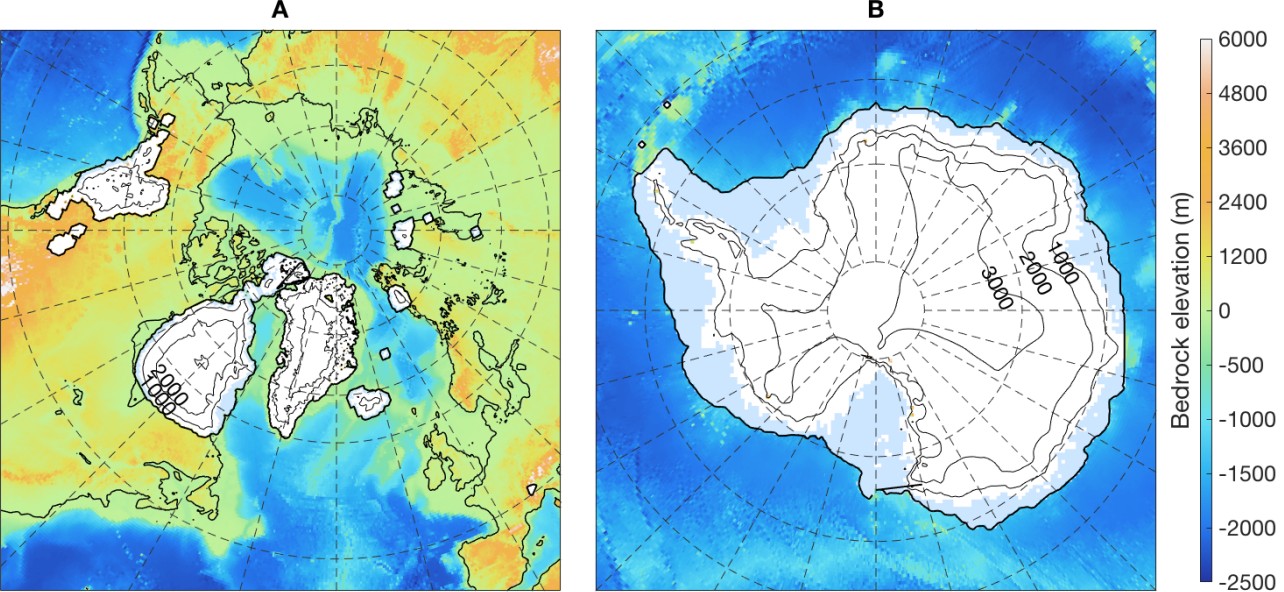

**Figure 10: The ice-sheets at the peak of MIS M2 (3.295 My ago), as simulated with the inverse-method forced matrix model. Contour lines for the Northern Hemisphere (A) show ice thickness, contour lines for Antarctica (B) show surface elevation. Antarctic ice shelves are shown as light blue. Bedrock elevation where not covered by ice is shown by colours. A sizeable ice sheet exists over the present-day Hudson Bay and Baffin Island, as well as a smaller one over the northern Cordillera. Antarctic ice volume increases by 1.5 – 3.5 m sea-level equivalent (SLE) because of the grounding of ice into the Filchner-Ronne basin.**

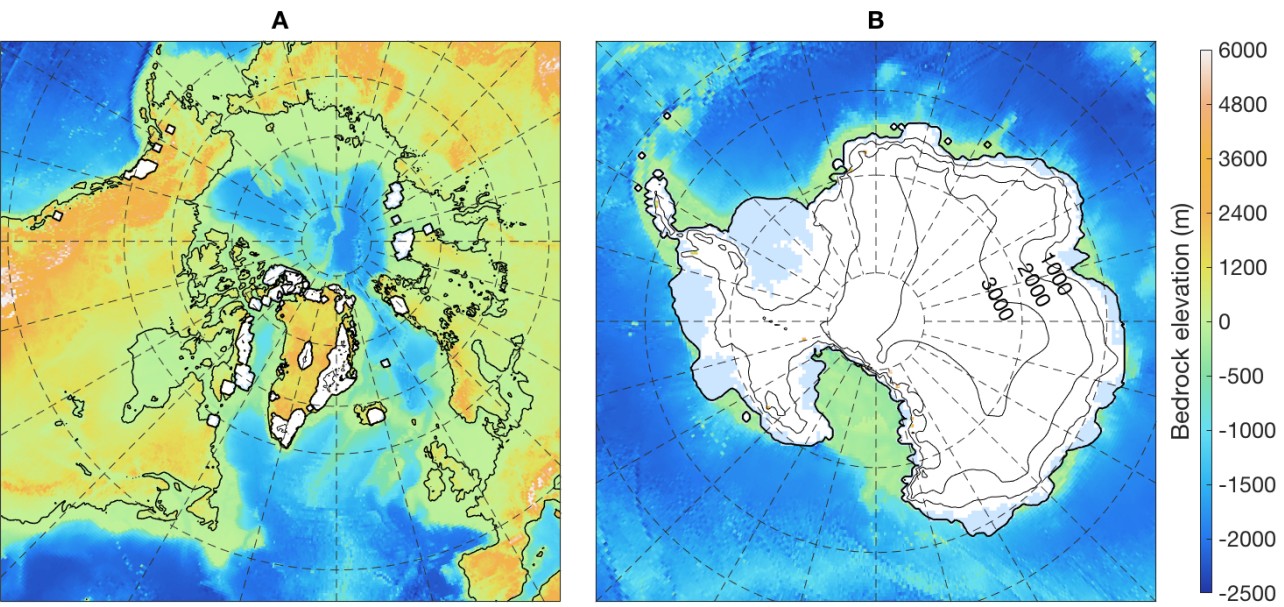

**Figure 11: The ice-sheets during KM5c (3.205 My ago), as simulated with the inverse-method forced matrix model. Contour lines for the Northern Hemisphere (A) show ice thickness, contour lines for Antarctica (B) show surface elevation. Antarctic ice shelves**

**are shown as light blue. Bedrock elevation where not covered by ice is shown by colours. Whereas most of the ice on Greenland has disappeared, retreat on Antarctica is limited to the Ross Sea, where the present-day ice shelf disintegrates to leave open ocean.**

Global mean sea level is compared to two different reconstructions in Fig. 12. Our model results generally lie between the $\delta^{18}O$-based reconstruction by Miller et al. (2011) and the reconstruction based on geological backstripping from New Zealand by Miller et al. (2012). During warm periods, our model generally shows lower sea levels and less variability than Miller et al. (2011) and Miller et al. (2012). During cold periods, our model generally shows less sea-level drop than the $\delta^{18}O$-based reconstruction by Miller et al. (2011), but more than the reconstruction based on geological backstripping. The $\delta^{18}O$-based reconstruction by Miller et al. (2011) is based on a linear relation between benthic $\delta^{18}O$ and sea level, which is an oversimplification of separating the contributions to the benthic $\delta^{18}O$ signal (e.g. Bintanja et al., 2005; de Boer et al., 2013). Miller et al. (2012) noted that reconstructing absolute values for local relative sea-level based on geological backstripping is difficult due to the required corrections for GIA and dynamic topography. However, the relatively short duration of MIS M2 means that the reconstructed drop in sea-level, of about 20 m relative to the background level, is likely to be accurate. Our model produces a value of about 24 m, in good agreement with this value.

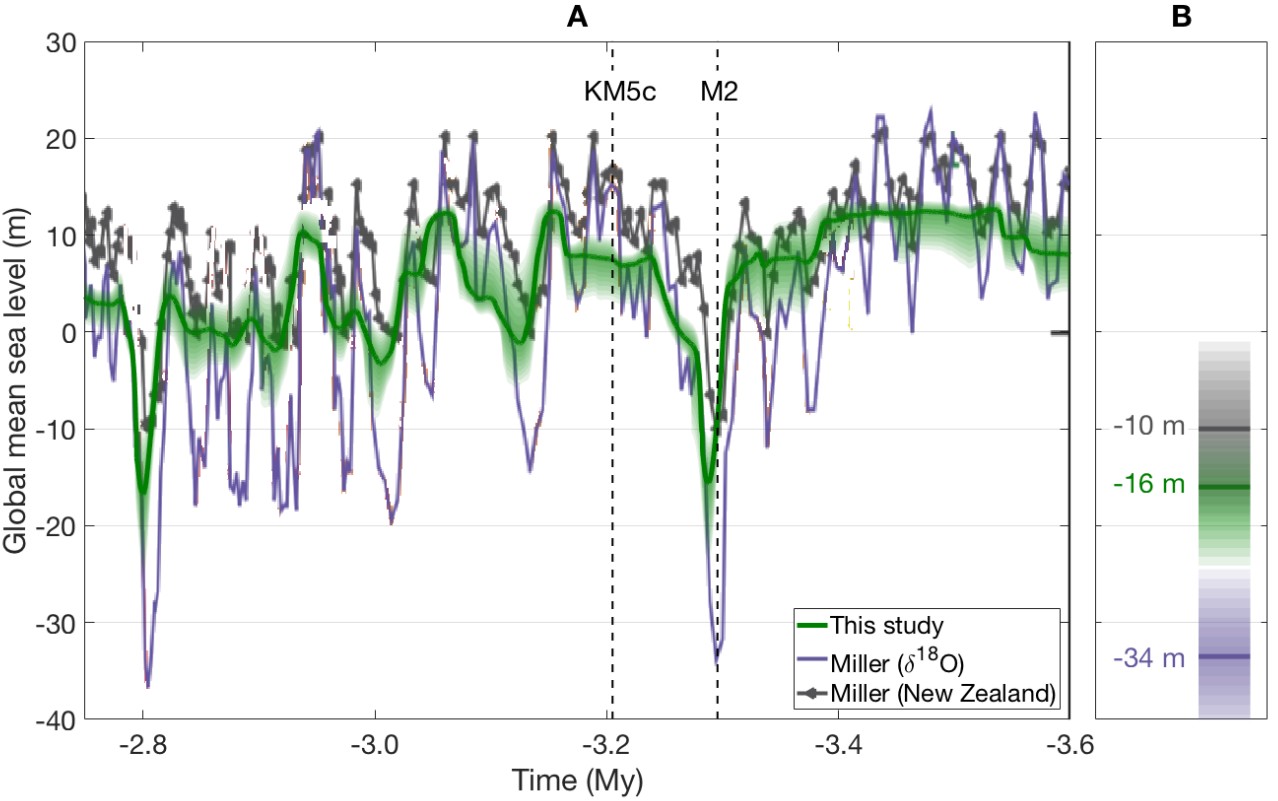

**Figure 12: A) Global mean sea level relative to the present day over time throughout the late Pliocene as simulated with the inverse-method forced matrix model, compared to reconstructions based on $\delta^{18}O$ (Miller et al., 2011; blue line) and geological backstripping (Miller et al., 2012; black line with triangles). Shaded areas show maximum uncertainty ranges. Vertical dashed lines indicate MIS**

**M2 and KM5c. B) Peak sea level drop during MIS M2 (3.3 My ago) for all three reconstructions, same vertical scale. Shaded areas show uncertainty ranges.**

The evolution of the West Antarctic ice sheet agrees partially with information derived from the AND-1B sediment core, recovered from beneath the northwest part of the Ross ice shelf by the ANDRILL programme (Naish et al., 2009; McKay et al., 2012). Information derived from this core by de Schepper et al. (2014) is compared to model results in Fig. 13. AND-1B shows ice-free conditions in the Ross Sea up to 3.4 Myr ago, followed by glacial deposits up to 3.24 Myr ago. Our model results show ice-free conditions up to 3.32 Myr ago, just prior to MIS M2. The ice-free conditions shown in our model results around KM5c cannot be validated by AND-1B due to a lack of data. Between 3.14 and 3.04 Myr ago, AND-1B again contains glacial deposits when our model results indicate ice-free conditions. The glacial conditions between 3.04 and 2.95 Myr ago and the subsequent ice-free conditions between 2.95 and 2.90 Myr ago indicated by AND-1B match with our model results. However, since the AND-1B sediment core contains several sizeable data gaps due to geological unconformities, the possibility that observed ~40 ky cycles in ice-rafted debris concentration have been incorrectly matched with 40 ky cycles in the $\delta^{18}$O age model can not be precluded. We therefore conclude that the AND-1B sediment core record can not be used to confirm or refute our model results.

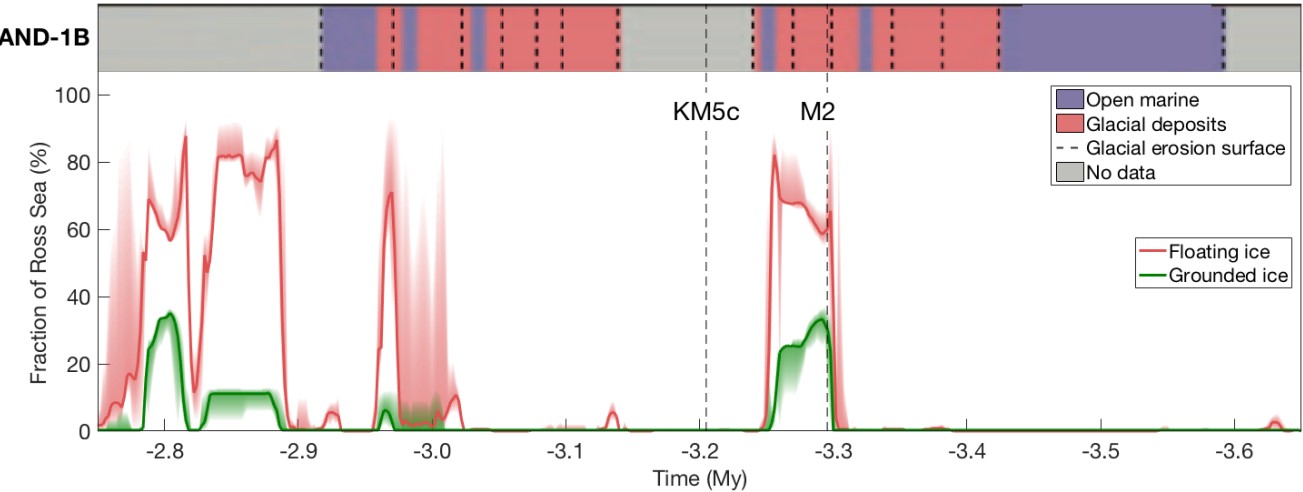

**Figure 13: Comparison of model results to the AND-1B sediment core (de Schepper et al., 2014). The top panel shows the glacial conditions derived from the sediment core, classified as either "open marine" (blue), "glacial deposits" (red) or "no data" (grey). The bottom panel shows the fraction of the Ross Sea that is covered by floating (red) and grounded (green) ice in our model simulations, with shaded areas showing the maximum uncertainty range from the sensitivity experiment.**

## 4 Discussion and conclusions

We have presented a new time-continuous, self-consistent reconstruction of pCO₂, ice sheet configuration and climate for the late Pliocene, 3.65 My – 2.75 My ago. Our approach is based on the matrix method by Berends et al. (2018), where an ice-sheet model is forced with a combination of several pre-calculated GCM snapshots. We have extended their two-state climate

matrix with several GCM snapshots created by Dolan et al. (2015), who simulated global climate during MIS M2 for different ice-sheet configurations and $pCO_2$ levels. Since our initial experiment, where this model was forced with the $pCO_2$ reconstruction by Stap et al. (2016), proved unable to constrain sea-level during MIS M2 any further, we adopted the inverse forward modelling approach by de Boer et al. (2013), forcing the model with the LR04 benthic $\delta^{18}O$ stack (Lisiecki and Raymo, 2005). By first using this $\delta^{18}O$-forced model set-up to simulate the last glacial cycle, we showed that it performed at least equally well to the $CO_2$-forced set-up by Berends et al. (2018) in terms of benthic $\delta^{18}O$ (Fig. 6) and surface temperature (Fig. 7), and better than the 1-D model set-up by Stap et al. (2016) in terms of simulated $pCO_2$ (Fig. 5).

Our results for the late Pliocene show a global mean sea-level drop of 10 – 25 m during MIS M2, with the uncertainty resulting from a sensitivity analysis investigating several key model parameters and the uncertainty in the applied $\delta^{18}O$ forcing. This value is in good agreement with the reconstruction based on geological backstripping from New Zealand by Miller et al. (2012; 10 ± 10 m) and the $\delta^{18}O$-based reconstruction by Miller et al. (2011; 34 ± 10 m). The extra ice with respect to present day is located mostly on eastern Canada and the northern Cordillera (9 – 20 m SLE) and the grounded ice over the Filcher-Ronne Sea (1.5 – 3.5 m SLE). The atmospheric $CO_2$ concentration necessary to produce the cooling required to grow these ice-sheets is shown to be 233 – 249 ppmv. During MIS KM5c, most of the ice on Greenland and West Antarctica disappears, raising global mean sea level to 3 – 10 m above present day, caused by a $pCO_2$ of 303 – 384 ppmv. The sea-level high stand of the mid-Pliocene Warm Period is achieved during MIS KM3 (3.155 My ago) at 8 – 14 m above present-day. The larger uncertainty in the modelled $pCO_2$ during warmer periods is attributed to the fact that the climate matrix used to force our ice sheet model contains only one GCM snapshot with a $pCO_2$ above present day levels, and only one ice sheet configuration with smaller than present ice sheets. Hence, the relationship between ice sheets and climate for warmer than present worlds is poorly constrained, which is reflected by an increased uncertainty in the simulated $pCO_2$ and ice volume. Expanding the climate matrix with additional GCM snapshots for intermediate $pCO_2$ levels, orbital configurations and ice-sheet geometries could help reduce this uncertainty, by more accurately capturing the non-linear response of many climatological parameters to these forcings and boundary conditions.

Despite the large uncertainty, our results suggest that $CO_2$ concentration during this warm time interval have not been significantly higher than present-day (~400 ppmv) values, in contrast to some of the proxy results. Comparing our Pliocene $pCO_2$ reconstruction to those by van de Wal et al. (2011) and Stap et al. (2016), our model shows stronger variability on the $10^4$ y timescale. In the long term, our model generally shows $pCO_2$ levels for warm climates that are higher than van de Wal et al. (2011) but lower than Stap et al. (2016). For colder climates, our $pCO_2$ is generally higher than Stap et al. (2016), and not clearly higher or lower than van de Wal et al. (2011). Given the level of disagreement between the different proxy-based reconstructions, it is not possible to assess the validity of the different model-based reconstructions relative to each other.

However, based on the ability of the different models to reproduce the EPICA $pCO_2$ record, assigning more confidence to the reconstruction presented here is justified.

Berends et al. (2018) provide a detailed discussion of the various advantages and disadvantages of the matrix method with respect to other methods of model forcing and coupling. Non-linear feedbacks of a growing ice sheet on the local and global climate, such as changes in atmospheric stationary waves, are not properly captured by this model set-up, although the inclusion of more GCM snapshots for intermediate-sized ice sheets should make the behaviour of the model more realistic in this respect. As a result the inception of the last glacial cycle (100 – 80 ky ago; Fig. 6, Fig. 7), is now also satisfyingly resolved in term of temperature and sea level drop though the decrease in $pCO_2$ seems stronger than suggested by the ice core record.

A drawback of the matrix method used here is that ocean temperature, required for calculating sub-shelf melt, is not included as a data field in the GCM snapshots. Instead, sub-shelf melt is calculated based on a combination of the temperature-based formulation by Martin et al. (2011) and the glacial–interglacial parameterization by Pollard and DeConto (2009), tuned by de Boer et al. (2013) to produce realistic present-day Antarctic shelves and grounding lines. Although Berends et al. (2018) show that this set-up performs well when simulating colder-than-present climates, this is not necessarily a priori true for warmer climates, where the ice shelves are expected to retreat or even disintegrate. A more elaborate parameterisation based on GCM-calculated ocean temperatures can be expected to produce more reliable results.

Similarly, the effect of changes in insolation upon surface temperature is not well constrained. The climate matrix proposed by Berends et al. (2018) uses a parameterisation based on the locally absorbed insolation. While this allows changes in prescribed insolation to affect climate by changing the relative weights assigned to the different GCM snapshots in the climate matrix, the different GCM snapshots used in the current version of the climate matrix were all forced with the same 3.3 Ma reconstruction by Lasker et al. (2004). Expanding the climate matrix with additional GCM snapshots for different orbital parameters, along the lines of Prescott et al. (2014; 2018), would make the relation between insolation and surface temperature more explicit. We believe this could possibly lead to a further retreat of the East Antarctic ice sheet during warm periods. Another possible hindrance to significant retreat of the Antarctic ice sheet in our simulations is the lack of explicit grounding line physics and relatively low model resolution, both of which have been shown to be required for accurate simulations of grounding line retreat (Schoof, 2007; Gladstone et al., 2012; Leguy et al., 2014). Instead, ANICE calculates sheet and shelf ice velocities using the SIA and SSA, respectively, and add these together, without additional grounding-line parameterisations.

An additional source of uncertainty in our reconstruction is the paleotopography of the period. Although we did include the Barents Sea erosion reversal by Butt et al. (2002) and its climate "fingerprint" as provided by Hill (2015) in our model, several other regions where ice may have existed during MIS M2 are suspected to have had a different topography - the Canadian

Archipelago has been suggested to have been still one unbroken landmass which only formed later through erosion by ice during the Pleistocene glaciations (Dowsett et al., 2016), the Hudson Bay was likely not yet submerged (present today mostly due to remaining isostatic depression from the Laurentide ice sheet (Dowsett et al., 2016; Raymo et al., 2011). Similarly, based on the Eocene-Oligocene Transition (34 My ago) paleotopography reconstruction by Wilson and Luyendyk (2009), it is

possible that, even during the Pliocene, West Antarctica was still mostly dry land (mostly submerged today due to erosion by ice and isostatic depression) and the Filchner-Ronne and Ross seas were significantly deeper (shallowed by ice-eroded sediment from West Antarctica). Although such changes in topography would likely have changed the evolution of the ice sheets, preliminary experiments for the Barents Sea showed that including the topography change without its GCM-calculated effect on climate resulted in a strong overestimation of ice volume, mostly because applying the present-day sea climate to the

newly exposed high-latitude landmass resulted in a strongly positive mass balance even with $pCO_2$ above 400 ppmv. Since no studies investigating the effects of these other topography changes on local and global climate are available yet, we did not include these changes in our study. Future work might be focused on reinvestigating these effects once results from new GCM simulations with these topography changes become available.

Considering the results from the comparison of our model output to the available proxy data and the different uncertainties and caveats in our results, we believe our results could be of added value to future iterations of the Pliocene Model Intercomparison Project (PlioMIP), to be used for example as boundary conditions for new GCM snapshots or even transient simulations.

*Code and data availability.* The reconstructed records of $pCO_2$, global mean sea level and benthic $\delta^{18}O$, as well as NetCDF files containing ice thickness, bedrock topography and annual mean surface temperature and precipitation for all four ice-sheet model regions during MIS M2, MIS KM5c and MIS KM3 are available online in the Supplement at https://doi.org/10.5281/zenodo.2598292 (Berends et al., 2019a).

*Author contributions.* CJB, BdB, and RSWvdW designed the study. AMD and DJH provided data from their own studies. CJB created the model set-up and carried out the simulations, with support from BdB and RSWvdW. CJB drafted the paper, and all authors contributed to the final version.

*Competing interests.* The authors declare that they have no conflict of interest.


*Acknowledgements.* The Ministry of Education, Culture and Science (OCW), in the Netherlands, provided financial support for this study via the program of the Netherlands Earth System Science Centre (NESSC). B. de Boer was funded by NWO Earth and Life Sciences (ALW), project 863.15.019. This work was sponsored by NWO Exact and Natural Sciences for the use of supercomputer facilities. Model runs were performed on the LISA Computer Cluster, we would like to acknowledge

SurfSARA Computing and Networking Services for their support. A.M. Dolan acknowledges funding from the European Research Council under the European Union's Seventh Framework Programme (FP7/2007–2013)/ERC grant agreement n° 278636.

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

**Table 1: Values of the different model parameters used in the LGC sensitivity analysis using the inverse-method forced matrix model. All model parameters were given upper and lower bounds 10 % above and below their benchmark value, except for the SIA/SSA flow enhancement factors (values based on Ma et al., 2010), the $\delta^{18}O$ forcing record (0.1 ‰ uncertainty stated by Lisiecki and Raymo, 2005) and the $pCO_2$ averaging time (values of 2,000 years and 15,000 years given by de Boer et al. (2014) and Stap et al. (2016), respectively).**

| Parameter | Description | Benchmark | Altered values |
| --- | --- | --- | --- |

| | | | | | | |
|---|---|---|---|---|---|---|
| $c_{abl,NAM}$ | *Ablation tuning parameter for North America (m/y)* | 0.189 | 0.173 | 0.205 | - | - |
| $c_{abl,EAS}$ | *Ablation tuning parameter for Eurasia (m/y)* | 0.256 | 0.233 | 0.282 | - | - |
| $c_{abl,GRL}$ | *Ablation tuning parameter for Greenland (m/y)* | 0.252 | 0.229 | 0.276 | - | - |
| $c_{abl,ANT}$ | *Ablation tuning parameter for Antarctica (m/y)* | 0.189 | 0.173 | 0.205 | - | - |
| $\delta^{18}O$ | *Benthic $\delta^{18}O$ forcing record* | LR04 | -0.1‰ | +0.1‰ | | |
| $e_{SIA}, e_{SSA}$ | *SIA/SSA flow enhancement factors* | 5.0, 0.5 | 4.5, 0.5 | 4.5, 0.7 | 5.6, 0.6 | 5.6, 0.7 |
| $r_{CO2}$ | *Ratio between $\delta^{18}O$ deviation and $pCO_2$ (Eq. 2)* | 120 | 108 | 132 | - | - |
| $r_{dT}$ | *Ratio between surface and deep-sea temperature anomaly* | 0.14 | 0.126 | 0.154 | - | - |
| $t_{CO2}$ | *$pCO_2$ averaging time in years (Eq. 2)* | 8,500 | 4,500 | 6,500 | 10,500 | 12,500 |

**Table 2: Sensitivity of the modelled $pCO_2$ and total eustatic sea-level contribution to the different model parameters at different points in the simulations.**

| Parameter | LGM | | MIS M2 - 3.3 My | | KM5c - 3.205 My | |
|---|---|---|---|---|---|---|
| | $pCO_2$ (ppmv) | Sea-level (m) | $pCO_2$ (ppmv) | Sea-level (m) | $pCO_2$ (ppmv) | Sea-level (m) |
| Benchmark | 192 | -98 | 242 | -16 | 319 | 7.4 |
| $c_{abl}$ | 188 – 197 | -87 – -95 | 241 – 244 | -15 – -17 | 317 – 328 | 4 – 8 |
| $\delta^{18}O$ | 191 – 194 | -86 – -90 | 233 – 249 | -10 – -25 | 303 – 384 | 3 – 8 |
| $e_{SIA}, e_{SSA}$ | 188 – 194 | -83 – -98 | 241 – 243 | -14 – -16 | 312 – 322 | 7 – 10 |
| $r_{CO2}$ | 190 – 196 | -88 – -98 | 242 – 243 | -15 – -16 | 319 – 322 | 7 – 7 |
| $r_{dT}$ | 194 – 194 | -87 – -93 | 240 – 245 | -14 – -17 | 317 – 323 | 7 – 8 |
| $t_{CO2}$ | 189 – 196 | -87 – -100 | 239 – 247 | -13 – -20 | 318 – 329 | 7 – 7 |
| Min - Max | 188 – 197 | -83 – -100 | 233 – 249 | -10 – -25 | 303 – 384 | 3 – 10 |