# Peer review of "Modelling ice sheet evolution and atmospheric CO2 during the Late Pliocene"

_Climate of the Past, 2019_

## Referee Comment (RC1) · Anonymous Referee #1 · 17 Apr 2019

The late Pliocene has been a focus for the paleoclimate data and modeling studies as its averaged warmer-than-present climate condition as well as the intensification of NH ice sheet occurred during this period. This paper presents the ice sheet and pCO2 evolution during the late Pliocene with the inverse modeling method and the asynchronous climate-ice sheet model approach. By combining validated modeling methods and Pliocene-based GCM climate matrix, the authors further constraint the pCO2 and sea level variations in particular for the MIS M2 and the late Pliocene warm interval. This work contributes importantly to the understanding of the climate variability and evolution for the late Pliocene in terms of transient modeling study. However, the current work is still not qualified for the final publication, I would recommend its publication after the following comments are considered.

[Figure]

General comments:

1. The introduction is too short to draw an overall background of this study. For example, concerning the late Pliocene warm period, the authors only list the related references without introducing the related results briefly. Much attention is paid on the MIS M2 and no introduction for the glacial interval after 3.0 Ma. However, the title of this paper indicates the objective of this study is to draw the ice sheet and pCO2 evolution over the late Pliocene and their transient simulation is also carried out from 3.65Ma to 2.75 Ma. Thus, the introduction needs to be modified or the title needs to be changed.

2. The authors validate the inverse method by applying it to investigate the last glacial cycle. In their results, the inversed pCO2 and modelled Benthic delta O18 show good agreement with the data. The modelled benthic delta O18 are largely improved comparing to their previous study (Berends et al., 2018), this is reasonable since the extra matrix provides more suitable climate states for the last glacial cycle. However, this extra climate matrix is not suitable for the late Pliocene. Unlike the PI, the pCO2 records during these warm periods are mostly higher than 300ppmv. In this climate matrix, there is only one warm state (PlioMIP,405ppmv) and it is far from the relative cold climate states, this will add more uncertainties to the warm period simulation for sure. To better understand the late Pliocene warm interval, at least, a medium warm-pCO2 (between 405 and 280 ppmv) and a strong-than-PI insolation climate matrix need to be included.

3. Please explain more details about the equation (2). What is the theoretical relation between (1) and (2) ? Why can this relation be also established during the late Pliocene without glacial-interglacial cycle?

Specific comments:

1. Line numbers are not continuous, it is not easy to comment.

2. Page 1 line 9: "such a climate state existed for a significant duration of time", please

specify how this climate state is.

3. In Figure 1: There are a lot of pCO2 data across this period, here the authors only show one inverse data which may mislead readers.

4. Page 2 line 9: "Over a period of about 20,000 years". Why is 20 kyrs, please provide the specific date for MIS M2.

5. Page 6 line 3: 200 ppmv, not 220 ppmv ?

6. Please describe the information for each labeled plot. Figure 9 is not labeled with the alphabet.

7. For the model parameters, the authors choose to vary the standard parameters by increasing or decreasing 10 %, does this value represent the range of parameter uncertainty ?

---

## Referee Comment (RC2) · Anonymous Referee #2 · 10 May 2019

The Pliocene is the focus of numerous modeling studies due to its warmer climate compared to present day. By understanding the climate variation of this period, we could tentatively understand the climatic consequences of our present-day warming climate. This paper uses a global coupled climate model with forcings including benthic oxygen observations along with inverse modeling method to model the evolutions of pCO2 and ice sheets. The authors have a particular focus on the MIS M2. This work shows promising results and contributes greatly to the field of paleoclimate modeling and could lead to a better understanding of our past climate and I would really like to see this work published. While the methodology is well thought of, other aspect of the paper requires consideration before being published.

General comments:

[Figure]

1. The paper is well written with good language and grammar.

2. The paper focuses a lot on the MIS M2 while the title mentions the modeling of the Pliocene.

3. Throughout the paper, acronyms and notations failed to be defined consistently as they appear in the text. I should note that the abstract should be considered somewhat independent from the main core of the paper and definitions and acronyms should be (re)defined starting from the introduction. Once they are defined they do not need to be redefined thereafter. Also, while some notation might be very common to some people specialized in the field, not everybody is an expert as they first read a paper.

4. Throughout the paper, please try to define/explain concepts, ideas or results directly when you mention them, not several paragraphs later. If a concept is explained later, simply indicate the section.

5. In general, the paper relies a lot on studies or modeling setup prior to this work. I am one these persons who does not like to have to read 10 papers before being able to know what is happening in one paper. I am not talking about adding in depth details but brief descriptions and summaries of the major model configuration or idea that is being used in the current study.

6. The introduction is relatively short for all the previous work it is trying to highlight. I would recommend briefly stating the findings of previous studies and contrast them with the novelty of the work presented here with obvious differences and the advantage(s) of your new approach. In particular, Berends et al 2018 is heavily cited throughout the paper and sometimes I felt like it was a prerequisite to this paper. A brief summary of this paper in the introduction would be appropriate.

7. Please provide more details regarding equations 1 and 2, their origins, citations, the link between the 2, and the definitions of the symbols. Also, are these equations valid for both the LGC and the Pliocene?

8. Many results should be emphasized quantitatively rather than using generic words such as "results agree well", . . .

9. The climate matrix is well suited for the validation of the LGC which presents colder climate compared to the Pliocene. I would say this is a major caveat of this study and ideally more Pliocene-like climates should be included in the matrix to validate the Pliocene.

10. The paper does not include a discussion of the importance of the perturbed parameters. Are any of them more sensitive than others? Along these lines, while parameters might have a big influence on model output, so does the resolution of the model particularly for ice sheets. A grid resolution of 40 km for Antarctica without proper grounding line dynamics sub-parameterization included in the dynamical solver will lead to large model differences (the same will hold true for Greenland but likely less so). I understand the computational expense of running paleo climate simulations, but the study would benefit from a run (even if it is for Antarctica only) at a resolution higher (20 km for Antarctica) of the one used in the current presentation and using the benchmark parameters. This run could be done for the Pliocene and stopped after MIS M2 and compare this portion of results only (to the 40 km with benchmark) if computing time is an issue.

11. The modeling section should include more details about the process of the modeling and clearly laying out the strengths and drawbacks. More importantly some parameters that are varied are not defined anywhere in the text (e.g. enhancement factor, . . .) They should be tied in with the modeling section with their impact on the model. Also, a better description (even brief) of the coupling should be included. Does the ice sheet model feeds back into the climate model in terms of ice sheet topography, ice extent and fresh water fluxes? Also, your model does include ocean forcing under floating ice which I discovered only at the end of the paper. This detail should be included in the modeling section when you are talking about the ice sheet model. Feel free to discuss the caveat of your method in the discussion.

Specific comments (I will highlight them as they appear in the paper). Some of the Rewordings are merely suggestions and you should feel free to consider them as you see fit.

1. Page 2 line 9: why focusing only on the first 20kyr and not the 60kyr that see the increase? Also, is the benthic oxygen time series by Lisiecky & Raymo the only one available? If not why use this one?

2. Page 2 line 8: define MIS M2 (you did in the abstract but please do it once in the main text).

3. Page 2 Figure 1: I am aware of the tendency in geological study to have the time series displayed with decreasing time but I still find it confusing, especially for modeling studies.

4. Page 3 line 4: add reference(s) for HadCM3.

5. Page 3 line 14: the matrix method is mentioned here for the first time but not explained. You will do so in section 2 so add something like "(see Sect. 2.3)".

6. Page 3 line 15: add reference(s) for ANICE.

7. Page 3 line 16: be more quantitative to define the word "accurately" (especially for paleo study).

8. Page 3 line 18: why is the matrix method applied specifically to MIS M2 while the title of the paper aims at modeling the Pliocene?

9. Page 3 line 25: define LR04 stack and add reference(s).

10. Page 3 line 27: replace "ice-climate" with "ice sheet-climate".

11. Page 4 line 7-8: "has resolution …". Maybe you could say a few words on the choice and practicality of this resolution.

12. Page 4 line 13: add reference(s) for SSA.

13. Page 4 line 14: add reference(s) for SIA.

14.Page 4 line 15: "with basal stress included in the SSA." On page 4 line 13 you mention that SSA is used only for floating ice and one can wonder why you include it in SSA. Instead add some precision in your text stating that there is a length scale over which the model transitions from SIA to SSA upward of ice shelves. This is what the PISM-PIK model does and your ice sheet model sounds to be very similar so I would add reference(s) to it.

15. Page 4 line 19: "for this application is 20 km for Greenland and 40 km for Antarctica...". Say a few words on the reason why you chose this coarse resolution for these ice sheets and why do you think they are adequate for the kind of modeling you are doing. (A recent paper (Goeltzer et al. 2016, Parameterization of basal friction near grounding lines in a one-dimensional ice sheet model) uses a resolution of 10 and 20 km for Greenland and Antarctica respectively.) Especially when modeling marine ice sheets, numerous studies have shown that a resolution of 100 m or higher is necessary to accurately model grounding line transition (Gladstone et al. 2012, Leguy et al. 2014, Parameterization of basal friction near grounding lines in a one-dimensional ice sheet model) which in turns impact sea level change prediction (which you are looking at in your paper). This requirement can be relaxed somewhat if using a grounding line parameterization which you are not. The lack of horizontal resolution for Greenland and Antarctica will likely lead to the greatest uncertainty in your study (even more so when applying basal melt rate under ice shelves). In the discussion, please clearly indicate what ongoing development you are considering for future modeling of the Pliocene.

16. Page 5 line 14: in a few words, describe what PRISM3 is, the configuration you are using, and reference(s).

17. Page 6 line 3: "200 ppmv pCO2", do you mean instead "220 ppmv"?

18. Page 6 line 9: "because the ICE-5G..." It is the first time you mention ICE-5G and you mention it later on (page 9) again. What is the difference between ICE-5G

ice sheets used in Pelletier 2004 and the ICE-5G LGM? Maybe a short paragraph about ICE-5G might be adequate, highlighting what is used for the purposed of your simulation and why.

19. Page 6 line 15: replace "Northern Hemisphere" by "Northern Hemisphere (NH)". The acronym "NH" is used two lines below without being defined.

20. Page 6 eq. 1: define the terms used in the equations and comment on the validity of the equation. Also, add reference(s).

21. Page 6 line 21: a simple suggestion, simply define the meaning of the upper bar notation in your equation this way you would not need to redefine both entities in each equation.

22. Page 6 line 22: replace "PD" by "present day (PD)".

23. Page 7 Figure 3: this figure is really nice, clear, and worth many words. Maybe it is worth mentioning earlier on, towards the beginning of Sect. 2.3.

24. Page 9 line 4: reword "last four glacial cycles" as "four last glacial cycles (LGC)". You mention the acronym LGC in table 1 without ever defining it.

25. Page 9 line 11: "The values that were used . . . Table 1." The text does no mention anything about the motivation of varying model parameters by 10%, nor in the table, please do so.

26. Page 10 line 8: please define the subscript "sw" somewhere.

27. Page 10 line 9: please define the subscript "dw" somewhere.

28. Page 10 line 15: replace "Surface temperature anomalies" by "Surface temperature anomalies ($\Delta T_s$)".

29. Page 10 line 15-16: divide the ice core records citation per ice sheet similarly to the way it is done in Figure 7 caption.

30. Page 10 Figure 6: the figures show results from Shakun et al. (2015) but these results are never referenced in the text. Please do so and describe their importance for your model comparison or do not display it. In the figure caption, be consistent with your label and add the publication year for Shakun et al. Also, explain what panels B-D are, not only A. Finaly, similarly to what is done in figure 5, provide some metric (like the Rˆ2 value) measuring the difference in fitting between this study and Berends et al. 2018.

31. Page 10 line 17: replace "temperature records is comparable..." by "temperature records are comparable". Also, be more quantitative rather than simply using "comparable". You can provide similar comparison as what you did for Figure 5 (using Rˆ2 and RMSE). For Antarctica one could argue that your results are not comparable especially between -80 kyr and -20 kyr. For Greenland, both models fail to capture the strong minimum around -70 kyr, is there an explanation for that?

32. Page 10 Figure 7: similar remark as for figure 6, provide quantitative metric.

33. Page 11 line 8: "is much better in the simulation here". Provide quantitative comparison.

34. Page 11 line 11: "possibly somewhat", This wording is really vague, try to omit it.

35. Page 11 line 9-13: "Whereas ..." this sentence carries a strong statement without being shown (you have not shown any sea-level variation for this study). Please provide a figure supporting your explanation. Also, this sentence is really long, try to split it in 2.

36. Page 11 line 16: "ice geometry". Nothing in the paper supports this claim. Either you have the results and you can show them or you could add something like "(not shown)".

37. Page 11 line 23: spell out GIA since used for the first time.

38. Page 11 section 3.2: add a few words in this section indicating why the climate

matrix you used to run your Pliocene simulation is appropriate.

39. Page 12 line 7: replace "sea-level drop" by "sea-level change". The values for KM5c are not dropping.

40. Page 12 line 7: "MIS M2 (3.295 My ago) . . .". No need to repeat "(3.295 My ago)", it should be mentioned earlier on.

41. Page 12-16: see general comment 9.

42. Page 12 line 18: replace "ice volume" by "sea-level contribution".

43. Page 12 line 20: ". . . when pCO2 rises again". A similar observation can be made with the strong peak at -2.8 My ago. Why is that time less important than MIS M2?

44. Page 13 Figure 9: mention in the caption what baseline you are using for your comparison (current sea level rise or from the beginning of the simulation?). On the figure, there is no need to repeat the y-axis titles for every subfigure after mentioning the one for North America. In your figure caption, please add something like "note the different y-axis limits". Also, replace "Volumes" (the first word in your caption) by "sea level contribution" to be consistent with your y-axis label.

45. Page 13 Figure 10: spell out SLE since never used before.

46. Page 14 line 6-8: which reconstruction should be used as a benchmark?

47. Page 14 line 6-7: "our model results agree well . . .". Similar to previous remarks, be more quantitative. Also, describe the way in which the results are in good agreement; is it because of the trend or else?

48. Page 15 Figure 12: similar to Figure 9, indicate your base or reference for the global mean sea level.

49. Page 15 line 14-16: "However, since . . .". I am puzzled with this sentence. It makes me believe that the AND-1B sediment core results cannot be trusted. I then wonder

why Figure 13 is shown at all in the paper and what value it adds. Maybe you could add this comparison (or lack of) to the discussion section or simply remove it from your paper.

50. Page 16 line 14-15: "it performed at least equally well". None of figures 5-7 show quantitative results supporting this statement. Figure 5 does not show results from Berends et al. 2018, and figure 6 A shows that this study performs better compared to Berends et al. 2018. Please reword.

51. Page 16 line 18: no need to repeat "(3.295 My ago)".

52. Page 16 line 23: "(1-3 m SLE)". Do you mean "(1.5-3.5 m SLE)" as stated in Fig. 10?

53. Page 17 line 1: no need to repeat "(3.205 My ago)".

54. Page 17 line 4-7: "The larger uncertainty . . .". Again, this is a strong drawback of this study. Please, discuss a bit further the impact on the Pliocene results or why the study is legitimate.

55. Page 17 line 19-24: some of the pros and cons of the matrix method mentioned here should be added in Sect. 2.3.

56. Page 17 line 27-28: "sub-shelf melt . . .". It would be good to add in Sect. 2.2 that sub-shelf melt rates are applied based on ocean forcing.

57. Page 17 line 27-28: "temperature-based formulation by Martin et al. (2011) . . .". Here or in the methodology section, be more specific on the relationship between melt rate and thermal forcing (i.e, linear like equation 5 in Martin et al., or quadratic as in Polard & DeConto (2012), or else). The impact could be important. Note that the relation suggested by Polard and DeConto is considered more accurate.

58. Page 17 line 32-33: ". . . and possibly . . .". I am struggling with this sentence as it sounds like it contradicts your sentence on page 15 line 16-17. Either the AND-1B

ice core has discrepancies and is not to be trusted or it should be and a more reliable comparison needs to be done in the result section.

59. Page 18 line 8: "East Antarctic . . . ". This is also true for West Antarctic ice sheet where grounding line dynamics plays an important role.

60. Page 18 line 10: "grounding line retreat (Schoof 2007,. . .)". Please, also cite Leguy et al. 2014 (Parameterization of basal friction near grounding lines in a one-dimensional ice sheet model) who went further in the grounding line influence as it investigated grounding line representation for different sliding laws, one being closer to the Mohr-Coulomb you are using in your model.

61. Page 18 line 26: Maybe replace "we can not . . ." by "we did not".

62. Page 24 Table 1: In column 4 (and sub-column 3) and row e_{SIA}, e_{SSA}, do you mean "5.6, 0.5"? (to be consistent with sub-column 1). If not why this difference in increment? Also, according to subsection 3.2, you not only used these model parameters for LGC but also for the Pliocene simulation.

---

## Author Comment (AC1) · 11 Jun 2019

Rebuttal to the review by Anonymous Referee 1

We thank the reviewer for their comments on the manuscript and would hereby like to address the concerns they raised. Comments in italics, below our rebuttal. Page and line numbers refer to the revised manuscript.

*The introduction is too short to draw an overall background of this study. For example, concerning the late Pliocene warm period, the authors only list the related references without introducing the related results briefly. Much attention is paid on the MIS M2 and no introduction for the glacial interval after 3.0 Ma. However, the title of this paper indicates the objective of this study is to draw the ice sheet and pCO2 evolution over*

*the late Pliocene and their transient simulation is also carried out from 3.65Ma to2.75 Ma. Thus, the introduction needs to be modified or the title needs to be changed.*

We agree that the introduction does not pay enough attention to the Pliocene other than MIS M2. We will expand this section. Since the results section presents and discusses results for the entire late Pliocene, focussing both on the cold M2 and the warm KM5c, we believe that, after expanding the introduction section to more adequately cover the entire late Pliocene, the title of the manuscript need not be changed.

**P2, L1-21: Briefly discussed the findings of referenced studies about the Pliocene climate, ice-sheets, sea-level and CO2.**

*The authors validate the inverse method by applying it to investigate the last glacial cycle. In their results, the inversed pCO2 and modelled Benthic delta O18 show good agreement with the data. The modelled benthic delta O18 are largely improved comparing to their previous study (Berends et al., 2018), this is reasonable since the extra matrix provides more suitable climate states for the last glacial cycle. However, this extra climate matrix is not suitable for the late Pliocene. Unlike the PI, the pCO2 records during these warm periods are mostly higher than 300ppmv. In this climate matrix, there is only one warm state (PlioMIP,405ppmv) and it is far from the relative cold climate states, this will add more uncertainties to the warm period simulation for sure. To better understand the late Pliocene warm interval, at least, a medium warm-pCO2(between 405 and 280 ppmv) and a strong-than-PI insolation climate matrix need to be included.*

While it is true that our climate matrix contains only one snapshot with CO2 higher than 280 ppmv, both the PRISM_280 and PRISM_220 snapshots are warmer than pre-industrial, due to the smaller ice sheets. This means that, even for the warmer-than-present eras, our climate matrix actually contains more information than the climate matrix used by Berends2018, which still managed to reproduce the last glacial cycle properly. We therefore believe our climate matrix is suitable for simulating the Pliocene.

While we agree with the reviewer that additional GCM snapshots for intermediate CO2 and insolation values and different ice sheets would be of added value, few such GCM simulations exist that are suitable for our study. A study that was recently published in GPC (Prescott et al. 2018: Regional climate and vegetation response to orbital forcing within the mid-Pliocene Warm Period: A study using HadCM3) could have provided useful data for us, but was only published after we'd already started this project. We agree that any new work on the Pliocene using our method could benefit from including these, and possible other, GCM results.

We will add a few lines to the manuscript describing this line of reasoning.

**P7, L10-14: Added a few lines justifying our use of the new, extended climate matrix for simulating the Pliocene.**

**P20, L18: Added a reference to Prescott2018 to the discussion.**

*Please explain more details about the equation (2). What is the theoretical relation between (1) and (2) ? Why can this relation be also established during the late Pliocene without glacial-interglacial cycle?*

Equation (1) quantifies the d18O-based inverse modelling method used by de Boer et al. (2013). Essentially, their model determines how global mean temperature (the inversely modelled variable) should have evolved, such that its combined effects on deep ocean temperature and global ice volume reproduce the observed d18O signal. Our own study takes this process one step further, by describing global climate not in terms of one single globally uniform temperature offset, but by using spatially variable temperature and precipitation fields. That way, our model determines how atmospheric CO2 should have evolved in order to change global climate in such a way that the resulting changes in ice volume and deep ocean temperature reproduce the observed d18O signal. Equation (2) quantifies how this is done; for every model time-step, the difference between the observed and modelled d18O signal is calculated. If the modelled value is not negative enough, this means that either the deep ocean is too warm,

or there is too little land ice. CO2 is then lowered for the next timestep, which leads to a cooling and therefore to more ice, bridging the gap between modelled and observed d18O. As long as there is a direct relation in the model between deep ocean temperature and pCO2, this method should produce accurate results even when there is little to no land ice present.

We agree that this conceptual description of the inverse modelling method should be included in the manuscript, and will add it to the Methodology section.

**P8, L13-19: Added a conceptual explanation of the inverse modelling method to the Methodology section.**

*Line numbers are not continuous, it is not easy to comment.*

We followed the Copernicus article template in restarting line numbers at 1 on every page.

*Page 1 line 9: "such a climate state existed for a significant duration of time", please specify how this climate state is.*

This means a warmer-than-present climate state.

**P1, L9: changed this.**

*In Figure 1: There are a lot of pCO2 data across this period, here the authors only show one inverse data which may mislead readers.*

We agree. We will add the available proxy records and the reconstruction by van de Wall 2011, which are used later on for model evaluation, to the figure.

**Figure 1: added CO2 proxy data and model reconstructions.**

*Page 2 line 9: "Over a period of about 20,000 years". Why is 20 kyrs, please provide the specific date for MIS M2.*

The warm peak in the d18O record directly prior to M2 occurs during MG1, at 3.315

My. The cold peak of M2 occurs about 20,000 years later, at 3.295 My. We will add this information to the manuscript.

**P3, L8: Added this information to the manuscript.**

*Page 6 line 3: 200 ppmv, not 220 ppmv?*

This is indeed a typo, we will fix it.

**P8, L1: Fixed the typo.**

*Please describe the information for each labeled plot. Figure 9 is not labeled with the alphabet.*

We agree that this should be fixed for all relevant figures in the manuscript. We will do so.

Figure 9: reordered the panels alphabetically.

Figure 10: added panel descriptions to the caption.

Figure 11: added panel descriptions to the caption.**Figure 6: added panel descriptions to the caption.**

**Figure 9: reordered the panels alphabetically.**

**Figure 10: added panel descriptions to the caption.**

**Figure 11: added panel descriptions to the caption.**
* * *
**Interactive comment on Clim. Past Discuss., https://doi.org/10.5194/cp-2019-34, 2019.**

---

## Author Comment (AC2) · 11 Jun 2019

Rebuttal to the review by Anonymous Referee 2

We thank the reviewer for their detailed comments on the manuscript, and would hereby like to address the concerns they raised. Comments in italics, below our rebuttal. Page and line numbers refer to the revised manuscript.

*The paper focuses a lot on the MIS M2 while the title mentions the modeling of the Pliocene.*

We agree with the reviewer on this point. We will add a more detailed discussion of earlier work on the Pliocene to the Introduction section of our manuscript. Since the results section presents and discusses results for the entire late Pliocene, focussing both

on the cold M2 and the warm KM5c, we believe that, after expanding the introduction section to more adequately cover the entire late Pliocene, the title of the manuscript need not be changed.

**P2, L1-21: Briefly discussed the findings of referenced studies about the Pliocene climate, ice-sheets, sea-level and CO2.**

*Throughout the paper, acronyms and notations failed to be defined consistently as they appear in the text. I should note that the abstract should be considered somewhat independent from the main core of the paper and definitions and acronyms should be(re)defined starting from the introduction. Once they are defined they do not need to be redefined thereafter. Also, while some notation might be very common to some people specialized in the field, not everybody is an expert as they first read a paper.*

*Throughout the paper, please try to define/explain concepts, ideas or results directly when you mention them, not several paragraphs later. If a concept is explained later, simply indicate the section.*

We agree that the manuscript was not always consistent in defining acronyms or field-specific concepts when they are first used. We have addressed all specific instances mentioned by the referee below, and corrected a few other ones we noticed ourselves.

*In general, the paper relies a lot on studies or modeling setup prior to this work. I am one these persons who does not like to have to read 10 papers before being able to know what is happening in one paper. I am not talking about adding in depth details but brief descriptions and summaries of the major model configuration or idea that is being used in the current study.*

*In particular, Berends et al 2018 is heavily cited throughout the paper and sometimes I felt like it was a prerequisite to this paper. A brief summary of this paper in the introduction would be appropriate.*

*The modeling section should include more details about the process of the modeling*

[Figure]

*and clearly laying out the strengths and drawbacks.*

*Also, a better description (even brief) of the coupling should be included. Does the ice sheet model feeds back into the climate model in terms of ice sheet topography, ice extent and fresh water fluxes?*

*Page 17 line 19-24: some of the pros and cons of the matrix method mentioned here should be added in Sect. 2.3. Page 6 line 15: replace "Northern Hemisphere" by "Northern Hemisphere (NH)". The acronym "NH" is used two lines below without being defined.*

*Page 6 eq. 1: define the terms used in the equations and comment on the validity of the equation. Also, add reference(s)*

*Page 6 line 21: a simple suggestion, simply define the meaning of the upper bar notation in your equation this way you would not need to redefine both entities in each equation*

*Page 6 line 22: replace "PD" by "present day (PD)"*

*Page 7 Figure 3: this figure is really nice, clear, and worth many words. Maybe it is worth mentioning earlier on, towards the beginning of Sect. 2.3.*

*Please provide more details regarding equations 1 and 2, their origins, citations, the link between the 2, and the definitions of the symbols. Also, are these equations valid for both the LGC and the Pliocene?*

We agree that the sections regarding the matrix method and the inverse modelling method were overly brief, and difficult to read for a reader not familiar with our earlier publications about these methods. We will rewrite and expand these parts of the Methodology section to make sure these methods can be understood in at least a qualitative sense without having to consult other publications.

**P6, L12-19: Expanded our conceptual explanation of the climate matrix method.**

**P8, L13 - P8, L19: Added a conceptual explanation of the inverse modelling method to the Methodology section.**

*The introduction is relatively short for all the previous work it is trying to highlight. I would recommend briefly stating the findings of previous studies and contrast them with the novelty of the work presented here with obvious differences and the advantage(s)of your new approach.*

We agree that the novelty of the approach to reconstructing CO2 in the way presented in our manuscript could be better illustrated in the introduction section. We will add a few lines to elaborate on this.

**P5, L1-4: Described how the d18O-based reconstructions by vdWal2011, Stap2016 and this one differ from other reconstructions.**

*Many results should be emphasized quantitatively rather than using generic words such as "results agree well",...*

We will discuss our results in a more quantitative sense where possible.

**Figure 6: added linear correlation coefficients between modelled and proxy-based d18O values.**

**P12, L22 – P13, L3: Added linear correlation coefficients and root mean square errors between modelled ice surface temperatures and ice core record.**

**Figure 7: added the same to the figure.**

**P17, L3-7: Changed the description of the comparison of our model results with the two Miller sea-level reconstructions to be more accurate.**

*The climate matrix is well suited for the validation of the LGC which presents colder climate compared to the Pliocene. I would say this is a major caveat of this study and ideally more Pliocene-like climates should be included in the matrix to validate the Pliocene.*

*Page 11 section 3.2: add a few words in this section indicating why the climate matrix you used to run your Pliocene simulation is appropriate.*

*Page 17 line 4-7: "The larger uncertainty...". Again, this is a strong drawback of this study. Please, discuss a bit further the impact on the Pliocene results or why the study is legitimate.*

While it is true that our climate matrix contains only one snapshot with $CO_2$ higher than 280 ppmv, both the PRISM_280 and PRISM_220 snapshots are warmer than pre-industrial, due to the smaller ice sheets. This means that, even for the warmer-than-present eras, our climate matrix actually contains more information than the climate matrix used by Berends2018, which still managed to reproduce the last glacial cycle properly. We therefore believe our climate matrix is suitable for simulating the Pliocene. While we agree with the reviewer that additional GCM snapshots for intermediate $CO_2$ and insolation values and different ice sheets would be of added value, few such GCM simulations exist that are suitable for our study. A study that was recently published in GPC (Prescott et al. 2018: Regional climate and vegetation response to orbital forcing within the mid-Pliocene Warm Period: A study using HadCM3) could have provided useful data for us, but was only published after we'd already started this project. We agree that any new work on the Pliocene using our method could benefit from including these, and possible other, GCM results. We will add a few lines to the manuscript describing this line of reasoning.

**P7, L10-14: Added a few lines justifying our use of the new, extended climate matrix for simulating the Pliocene.**

**P20, L18: Added a reference to Prescott2018 to the discussion.**

*The paper does not include a discussion of the importance of the perturbed parameters. Are any of them more sensitive than others?*

*Page 9 line 11: "The values that were used...Table 1." The text does no mention*

[Figure]

*anything about the motivation of varying model parameters by 10%, nor in the table, please do so.*

For the benthic d18O forcing, the investigate range of values is based on the uncertainty reported by Lisiecki and Raymo, 2005. For the SIA/SSA flow enhancement factors, the range of allowed values is based on the results reported by Ma et al., 2010. For the CO2 averaging time, the range of tested values is based on the values used by de Boer et al., 2013 and Stap et al., 2016. The optimum ratio between surface and deep-water temperature anomalies, and the ratio between d18O and CO2 changes, were determined experimentally by de Boer et al. (2010), building on earlier work by Bintanja and van de Wal (2008). While de Boer et al. (2010) investigated a wider range of +-25% for these parameters, their aim was to test the robustness of the inverse modelling method itself. Based on their results, and those of similar studies (de Boer et a., 2014; Stap et al., 2016), we assumed this method to be proven robust, and consequently adopted more conservative values. For the ablation tuning parameters, we used a range of +- 10% based on our earlier work (Berends et al., 2018), where we found that this range resulted in an uncertainty in modelled sea-level equivalent ice volumes at LGM that matched the uncertainty in other reconstructions. We agree that this information should be provided in the manuscript. The sensitivity of the model results (modelled pCO2 and sea-level at LGM, M2 and KM5c) to all these parameters is listed in Table 1, which is referred to in section 3.1 and section 3.2.

**P11, L12-19: Added this information to the manuscript.**

*Along these lines, while parameters might have a big influence on model output, so does the resolution of the model particularly for ice sheets. A grid resolution of 40 km for Antarctica without proper grounding line dynamics sub-parameterization included in the dynamical solver will lead to large model differences (the same will hold true for Greenland but likely less so). I under-stand the computational expense of running paleo climate simulations, but the study would benefit from a run (even if it is for Antarctica only) at a resolution higher (20 km for Antarctica) of the one used in the current*

*presentation and using the benchmark parameters. This run could be done for the Pliocene and stopped after MIS M2 and compare this portion of results only (to the 40 km with benchmark) if computing time is an issue.*

*Page 4 line 15: "with basal stress included in the SSA." On page 4 line 13 you mention that SSA is used only for floating ice and one can wonder why you include it in SSA. Instead add some precision in your text stating that there is a length scale over which the model transitions from SIA to SSA upward of ice shelves. This is what the PISM-PIK model does and your ice sheet model sounds to be very similar so I would add reference(s) to it.*

*Page 4 line 19: "for this application is 20 km for Greenland and 40 km for Antarctica...". Say a few words on the reason why you chose this coarse resolution for these ice sheets and why do you think they are adequate for the kind of modeling you are doing. (A recent paper (Goeltzer et al. 2016, Parameterization of basal friction near grounding lines in a one-dimensional ice sheet model) uses a resolution of 10 and 20km for Greenland and Antarctica respectively.) Especially when modeling marine ice sheets, numerous studies have shown that a resolution of 100 m or higher is necessary to accurately model grounding line transition (Gladstone et al. 2012, Leguy et al. 2014, Parameterization of basal friction near grounding lines in a one-dimensional ice sheet model) which in turns impact sea level change prediction (which you are looking at in your paper). This requirement can be relaxed somewhat if using a grounding line parameterization which you are not. The lack of horizontal resolution for Greenland and Antarctica will likely lead to the greatest uncertainty in your study (even more so when applying basal melt rate under ice shelves). In the discussion, please clearly indicate what ongoing development you are considering for future modeling of the Pliocene.*

The migration of grounding lines and their effect on ice sheet dynamics, resulting in the proper glacial-interglacial changes in Antarctic ice sheet volume, are captured in ANICE by using the combination of both SIA and SSA velocities in the transition zone (over land, the SIA and SSA velocities are summed, resulting in a smooth transition).

In earlier, published work, we already showed that the combination of SSA and SIA in the transition zone works well for glacial-interglacial cycles (De Boer et al., 2013, based also on the work of Martin and Winkelmann with PISM). Our model uses this approach, and we showed in our Berends et al. 2018 paper that the 15 m sea-level contribution from Antarctica at LGM (and the subsequent retreat to present-day geometry) is reproduced by the model. Although the manuscript doesn't include a figure for the volumes of the four ice-sheets in the 410 ky benchmark experiment, we find the same results with the current model set-up. We will add a few lines to Section 2.2 (ice-sheet model) to describe this combined SIA/SSA approach and how it reproduces Antarctic glacial-interglacial dynamics.

**P5, L21-24: Added a few lines describing the way SIA and SSA velocities are combined, and how this approach reproduces Antarctic grounding line migration and glacial-interglacial ice volume changes.**

While we agree with the reviewer that a 20 km run for Antarctic could be of interest, several parameterisations in both the ice model and the matrix forcing are tuned for the 40 km version, and would require retuning for a 20 km simulation, which is a time-consuming endeavour. The current model version simulates about 150,000 model years in a day. Doubling the resolution makes the model about 8 times slower, which would make it more complicated to use on the computing facilities available to us. We believe that the added value of such an experiment currently does not outweigh the required effort, particularly not because there is no clear argument why enhancing the resolution of a model that already reproduces grounding line migration would influence the fundamental behaviour of the model.

*Some parameters that are varied are not defined anywhere in the text (e.g. enhancement factor,..). They should be tied in with the modeling section with their impact on the model.*

We will add a description of the function of these two parameters to the manuscript.

**P11 L13-15: Added a description of the meaning of the SIA/SSA flow enhancement factors.**

*Also, your model does include ocean forcing under floating ice which I discovered only at the end of the paper. This detail should be included in the modeling section when you are talking about the ice sheet model.*

*Page 17 line 27-28: "sub-shelf melt...". It would be good to add in Sect. 2.2 that sub-shelf melt rates are applied based on ocean forcing.*

*Page 17 line 27-28: "temperature-based formulation by Martin et al. (2011) ...". Here or in the methodology section, be more specific on the relationship between melt rate and thermal forcing (i.e., linear like equation 5 in Martin et al., or quadratic as in Polard DeConto (2012), or else). The impact could be important. Note that the relation suggested by Polard and DeConto is considered more accurate*

Basal melt is parameterised in our model using the linear relation between ocean temperature based on Pollard and DeConto (2009) and Martin (2011), including the parameterisation of sub-shelf cavity circulation based on the shortest linear distance the open ocean from Pollard and DeConto (2009). Both Pollard and DeConto (2009) and de Boer et al. (2013) show that this approach yields reasonable present-day Antarctic ice shelves. A recent publication by Lazeroms et al. (2018) shows that, although the quadratic formulation by Pollard and DeConto (2012) does result in slightly more accurate average melt rates, both the linear and quadratic formulations fail to accurately resolve variance within basins, which they successfully simulate using their plume parameterisation. We therefore believe that the next step in representing basal melt in our model would not be the quadratic formulation, but rather a more detailed plume parameterisation along the lines of Lazeroms et al. (2018). We agree that this line of reasoning should be added to the Methodology section of the manuscript.

**P5, L30 – P6, L5: Added a description of the way sub-shelf melt is calculated in the model.**

*Page 2 line 9: why focusing only on the first 20kyr and not the 60kyr that see the increase? Also, is the benthic oxygen time series by Lisiecky Raymo the only one available? If not why use this one?*

We agree that, in order to illustrate both the magnitude and duration of the M2 glacial event, both the duration of the inception and the termination are important. We will change this line to reflect this. Because we want to investigate global land ice volume, a stack of globally distributed benthic d18O records is required. A few different options are available (Imbrie et al., 1984; Lisiecki and Raymo, 2005; Zachos et al., 2001, 2008; Cramer et al., 2009). Since all of these are based on mostly the same data for the first few million years, the differences between them are very small. The choice to use LR04 was mainly motivated by a desire for consistency with other inverse-routine based reconstructions (e.g. van de Wal et al., 2011; de Boer et al., 2013; Stap et al., 2016). We will explain this choice in the manuscript.

**P9, L5-9: Added a few lines motivating the choice of LR04 as forcing.**

*Page 2 line 8: define MIS M2 (you did in the abstract but please do it once in the main text).*

We will do so.

**P3, L7: Provided the meaning of the abbreviation "MIS"**

*Page 2 Figure 1: I am aware of the tendency in geological study to have the timeseries displayed with decreasing time but I still find it confusing, especially for modelling studies.*

While we sympathise with the reviewer's confusion, we follow the tendency of the majority of paleo-modelling studies to display time in the "geological", right-to-left fashion.

*Page 3 line 4: add reference(s) for HadCM3.*

We will do so.

**P5, L7: added two references to studies describing the original and most recent versions HadCM3**

*Page 3 line 14: the matrix method is mentioned here for the first time but not explained. You will do so in section 2 so add something like "(see Sect. 2.3)".*

We will do so.

**P4, L15: referred to Section 2.3 for a description of the matrix method of model coupling**

*Page 3 line 15: add reference(s) for ANICE.*

We will do so.

**P4, L15-16: added references to studies describing ANICE, as well as to Section 2.2 of this manuscript.**

*Page 3 line 16: be more quantitative to define the word "accurately" (especially for paleo study).*

We will do so.

**P4, L19-20: described which parameters (ice volume, deep-water temperature, benthic ïĄd'18O, etc.) were accurately reproduced by Berends et al. (2018)**

*Page 3 line 18: why is the matrix method applied specifically to MIS M2 while the title of the paper aims at modeling the Pliocene?*

It was applied to the whole Late Pliocene. We will fix this line.

**P4, L20: changed "MIS M2" to "the late Pliocene"**

*Page 3 line 25: define LR04 stack and add reference(s).*

LR04 should have been defined earlier. We will fix this.

**P3, L8: defined LR04**

**Figure 1: defined LR04 in the caption**

**P4, L28: added a reference for LR04**

*Page 3 line 27: replace "ice-climate" with "ice sheet-climate".*

We will do so.

**P4, L30: fixed this.**

*Page 4 line 7-8: "has resolution...". Maybe you could say a few words on the choice and practicality of this resolution.*

We never actually ran HadCM3 ourselves – the GCM simulations that provided the data used in our study were performed by Singarayer and Valdes (2010) and Dolan et al. (2015). We hope the extra information we added to Section 2.3 about the way the matrix method works will resolve any confusion in this regard.

*Page 4 line 13: add reference(s) for SSA. Page 4 line 14: add reference(s) for SIA*

We will do so.

**P5, L17: added a reference for the SSA**

**P5, L18: added a reference for the SIA**

*Page 6 line 3: "200 ppmv pCO2", do you mean instead "220 ppmv"?*

The reviewer is correct. We will fix this.

**P8, L1: Fixed this.**

*Page 5 line 14: in a few words, describe what PRISM3 is, the configuration you are using, and reference(s)*

*Page 6 line 9: "because the ICE-5G..." It is the first time you mention ICE-5G and you mention it later on (page 9) again. What is the difference between ICE-5G ice sheets used in Pelletier 2004 and the ICE-5G LGM? Maybe a short paragraph about ICE-5G*

*might be adequate, highlighting what is used for the purposed of your simulation and why.*

ICE-5G is a reconstruction of the evolution of all ice on Earth, as well as the vertical motion of the solid Earth, from 21 ky ago to the present day. Dolan et al. (2015), whose GCM simulation results we use to force our ice-sheet model, chose to use two time slices of ICE-5G as boundary conditions for their GCM simulations. We will clarify this in the manuscript.

**P7, L2-5: added a brief description of the PRISM3 and other ice-sheet configurations, with references.**

*Page 9 line 4: reword "last four glacial cycles" as "four last glacial cycles (LGC)". You mention the acronym LGC in table 1 without ever defining it.*

The abbreviation "LGC" in the header of Table 1 was erroneous, since the listed parameter values were used for both the simulations of the last four glacial cycles and of the Pliocene. We will fix this.

**Table 1: removed "LGC" from the header.**

*Page 10 line 8: please define the subscript "sw" somewhere.*

*Page 10 line 9: please define the subscript "dw" somewhere.*

We will do so.

**Figure 6: added definitions to the figure caption.**

*Page 10 line 15: replace "Surface temperature anomalies" by "Surface temperature anomalies $(\Delta T\_s)$".*

*We will do so.*

***Figure 7: added definition to the figure caption.***

*Page 10 line 15-16: divide the ice core records citation per ice sheet similarly to the*

*way it is done in Figure 7 caption.*

*We will do so.*

**P12, L22 – P13, L3: fixed the references to the ice core temperature records**

*Page 10 Figure 6: the figures show results from Shakun et al. (2015) but these results are never referenced in the text. Please do so and describe their importance for your model comparison or do not display it. In the figure caption, be consistent with your label and add the publication year for Shakun et al. Also, explain what panels B-D are, not only A. Finaly, similarly to what is done in figure 5, provide some metric (like the RËĘ2 value) measuring the difference in fitting between this study and Berends et al.2018.*

*We will add a description of the Shakun et al. data to the text, as well as a reference. We will fix the figure.*

**P12, L9-10: added description of and reference to Shakun 2015**

**Figure 6: added proper descriptions of the panels, fixed the missing year of publication of Shakun 2015, and added R2 values.**

*Page 10 line 17: replace "temperature records is comparable..." by "temperature records are comparable". Also, be more quantitative rather than simply using "comparable". You can provide similar comparison as what you did for Figure 5 (using RËĘ2 and RMSE). For Antarctica one could argue that your results are not comparable especially between -80 kyr and -20 kyr. For Greenland, both models fail to capture the strong minimum around -70 kyr, is there an explanation for that?*

*Page 10 Figure 7: similar remark as for figure 6, provide quantitative metric*

*We agree that the comparison between modelled ice surface temperatures and proxy-based reconstructions of the same should be more quantitative. We will add R2 and RMSE values for both model versions and both records. The minimum in reconstructed*

*surface temperatures on both ice-sheets around 70 kyr ago is not as pronounced in the d18O record, so we do not expect to see it in the model results. Why this minimum is visible in both ice core records but not in the benthic records is unclear, but it is outside of the scope of this study.*

*P13, L4–7: Added linear correlation coefficients and root mean square errors between modelled ice surface temperatures and ice core record.*

**Figure 7: added the same to the figure.**

*Page 11 line 8: "is much better in the simulation here". Provide quantitative comparison.*

*We will do so.*

**P13, L17-20: Provided R2 for both simulations between 120 and 80 kyr.**

*Page 11 line 11: "possibly somewhat", This wording is really vague, try to omit it.*

*We agree. We will fix this.*

**P13, L22: removed "possibly somewhat"**

*Page 11 line 9-13: "Whereas..." this sentence carries a strong statement without being shown (you have not shown any sea-level variation for this study). Please provide a figure supporting your explanation. Also, this sentence is really long, try to split it in 2.*

*The sentence should have read "benthic d18O contributions" instead of "sea-level" (which is shown). We will fix this. We will also split the long sentence into two shorter ones.*

**P13, L21-23: fixed the line.**

*Page 11 line 16: "ice geometry". Nothing in the paper supports this claim. Either you have the results and you can show them or you could add something like "(not shown)".*

*The reviewer is correct – we do indeed have the comparison, but we did not show it*

*in order to keep the number of Figures manageable. We will add "(not shown)" to the line.*

**P14, L3: added this.**

*Page 11 line 23: spell out GIA since used for the first time.*

*We will do so.*

**P14, L10-11: did so.**

*Page 12 line 7: replace "sea-level drop" by "sea-level change". The values forKM5c are not dropping.*

*We will do so.*

**P14, L24: changed "sea-level drop" to "sea-level change".**

*Page 12 line 7: "MIS M2 (3.295 My ago)...". No need to repeat "(3.295 My ago)",it should be mentioned earlier on.*

*We will change this.*

**P14, L24: removed time of MIS M2**

*Page 12 line 18: replace "ice volume" by "sea-level contribution".*

*We will change this.*

**P15, L8: changed this sentence.**

*Page 12 line 20: "...when pCO2 rises again". A similar observation can be made with the strong peak at -2.8 My ago. Why is that time less important than MIS M2?*

*The strong peak at 2.8 My represents the first of the Pleistocene glaciations. While this lies outside of the focus of this study, we agree that, since it appears in the results, it should be verbally described. We will add a line to the manuscript.*

**P15, L16-18: Added a few lines describing the simulated onset of the Pleistocene glaciations.**

*Page 13 Figure 9: mention in the caption what baseline you are using for your comparison (current sea level rise or from the beginning of the simulation?). On the figure, there is no need to repeat the y-axis titles for every subfigure after mentioning the one for North America. In your figure caption, please add something like "note the different y-axis limits". Also, replace "Volumes" (the first word in your caption) by "sea level contribution" to be consistent with your y-axis label.*

*The figure shows sea-level contribution relative to present-day; a value of 0 for Greenland means it has an ice-sheet of the same size as the present-day one, whereas a value of 7.4 means a completely ice-free island. The curves don't start at zero because, as is stated in the first paragraph of Section 3.2, the model is initialised with the PRISM3 ice sheets. We will add this information to the figure caption. We will also remove the redundant Y axis labels.*

**Figure 9: removed repeating Y axis labels, updated caption.**

*Page 13 Figure 10: spell out SLE since never used before.*

*We will do so.*

**Figure 10: did so.**

*Page 14 line 6-8: which reconstruction should be used as a benchmark?*

*The d18O-based reconstruction by Miller et al. (2011) uses a linear relation between d18O and sea level, which studies like Bintanja et al. (2005) and de Boer et al. (2013) have shown to be an oversimplification. The reconstruction by Miller et al. (2012) based on geological backstripping has been noted to be difficult to express in absolute terms, which require accurate corrections for dynamic topography. However, relative changes over shorter periods should be more accurate. We will discuss this in the manuscript.*

***P17, L7-12: added a brief discussion of the merits of both sea-level reconstructions.***

*Page 14 line 6-7: "our model results agree well...". Similar to previous remarks, be more quantitative. Also, describe the way in which the results are in good agreement; is it because of the trend or else?*

*The uncertainty in our results overlaps with the uncertainty in the reconstructions by Miller 2011 and 2012. We agree that the phrase "results agree well" is too vague. We will change this.*

***P17, L3-7: changed this sentence.***

*Page 15 Figure 12: similar to Figure 9, indicate your base or reference for the global mean sea level.*

*As before, the reference is the present day. We will state this in the figure caption.*

***Figure 12: did so.***

*Page 15 line 14-16: "However, since...". I am puzzled with this sentence. It makes me believe that the AND-1B sediment core results cannot be trusted. I then wonder why Figure 13 is shown at all in the paper and what value it adds. Maybe you could add this comparison (or lack of) to the discussion section or simply remove it from your paper.*

*We agree that the comparison between our model results and the AND-1B sediment core does not provide any new insights. However, when discussing our findings informally with acquainted geologists, the Antarctic sediment cores were mentioned multiple times as a possibly helpful source of information. We therefore believe that we should still show the comparison, and explain it is not helpful.*

*Page 16 line 14-15: "it performed at least equally well". None of figures 5-7 show quantitative results supporting this statement. Figure 5 does not show results from Berends et al. 2018, and figure 6 A shows that this study performs better compared to*

*Berends et al. 2018. Please reword.*

*In response to earlier comments by the referee, Figures 6 and 7 have now been updated to show a quantitative assessment of the performance of both models (linear correlation coefficients between modelled d18O and temperatures, and proxy reconstructions of these parameters). The statement that our model performs better than the earlier version by Berends et al. (2018) is now justified. Figure 5 showed that our model performed better than the one by Stap et al, 2016., not the one by Berends et al., 2018. We will rectify this.*

**P19, L5: changed the sentence to more accurately describe the comparison between different model performances.**

*Page 16 line 18: no need to repeat "(3.295 My ago)".*

*We will fix this.*

**P19, L6: fixed this.**

*Page 16 line 23: "(1-3 m SLE)". Do you mean "(1.5-3.5 m SLE)" as stated in Fig.10?*

*We do indeed. We will fix this.*

**P19, L12: fixed this.**

*Page 17 line 1: no need to repeat "(3.205 My ago)".*

*We will fix this.*

**P19, L13: fixed this.**

*Page 17 line 32-33: "...and possibly...". I am struggling with this sentence as it sounds like it contradicts your sentence on page 15 line 16-17. Either the AND-1B ice core has discrepancies and is not to be trusted or it should be and a more reliable comparison needs to be done in the result section.*

*We agree that, since we already concluded that the AND-1B sediment core cannot be*

*used for model validation, this sentence should be removed.*

**P20, L11: removed this sentence.**

*Page 18 line 8: "East Antarctic...". This is also true for West Antarctic ice sheet where grounding line dynamics plays an important role.*

*We agree with the reviewer that a proper description of grounding line dynamics in the model could also affect our results for West Antarctica during intermediately warm periods. We will change the sentence to accurately describe this.*

**P20, L20: changed the line to more properly describe the caveat of missing grounding line physics in our model**

*Page 18 line 10: "grounding line retreat (Schoof 2007, ...)". Please, also cite Leguy et al. 2014 (Parameterization of basal friction near grounding lines in a one-dimensional ice sheet model) who went further in the grounding line influence as it investigated grounding line representation for different sliding laws, one being closer to the Mohr-Coulomb you are using in your model.*

*We agree with the reviewer that this is a valuable study to cite in our manuscript. We will do so.*

**P20, L22: added a reference to Leguy et al. (2014) to the manuscript.**

*Page 18 line 26: Maybe replace "we can not..." by "we did not".*

*We will do so.*

**P21, L5: did so**

*Page 24 Table 1: In column 4 (and sub-column 3) and row e_SIA, e_SSA, do you mean "5.6, 0.5"? (to be consistent with sub-column 1). If not why this difference in increment? Also, according to subsection 3.2, you not only used these model parameters for LGC but also for the Pliocene simulation.*

*We did indeed use these parameters for all simulations, and fixed the table header in response to an earlier comment by the reviewer. Regarding the values for the SIA and SSA flow enhancement factors: Ma et al. (2010), whose results we used to choose these values, provide upper and lower bounds for both parameters, as well for the ratio between the two. This means that not all parameter value pairs within the ranges for each individual parameter are allowed. According to them, e_SIA may only be 10 times larger than e_SSA. The upper bound of 5.6 for e_SIA is therefore only allowed when e_SSA is larger than 0.56, which rounds off to 0.6.*
* * *
*Interactive comment on Clim. Past Discuss., https://doi.org/10.5194/cp-2019-34, 2019.*

---

## Author Response (AR1)

**Author's comment replying to editor's comments**

We thank the editor for reviewing our rebuttals, and would hereby like to address the concerns raised.

Comments in italics, below our rebuttal. Page and line numbers refer to the revised manuscript.

*1) Both reviewers pointed out that the introduction is insufficient. Please do extend the introduction to cover recent publications such as Prescott et al., (2018), Tan et al., (2017, EPSL), etc.*

We agree with the editor and the reviewers that the introduction did not give an adequate description of the current state of research into the Pliocene in general, but focused too much on just MIS M2. We have expanded the introduction with a broader overview of the Pliocene, both from a model- and a data-based perspective. We have also added a few lines describing how our own work fits into this broader picture.

*2) Both reviewers mentioned the lack of description of model configuration, and they are both confused by equation 1 and 2 regarding how it is implemented and how the scaling factor is determined in 2. Those are valid and important points to address. From what I understand, it looks like, in both equation 1 and 2, (t+0.1ky) should be a subscript instead of an actual term. If so, please correct.*

We agree with the editor and the reviewers that the manuscript lacked a proper conceptual explanation of the inverse method, relying too much on the reader's supposed familiarity with our earlier work. We revised the section explaining this method, adding a brief conceptual explanation of what exactly this method does, and how it is quantified by the two equations. The (t+0.1ky) in both equations is a function argument (e.g. d18O_obs(t+0.1ky) is "d18O 100 years after the current model time"). We believe that the revised text resolves the confusion about this term.

*3) Both reviewers pointed out that a major caveat of this study is the lack of warm Pliocene-like climates in the climate matrix. Given that vegetation, ocean circulation, and ice sheet all respond nonlinearly to insolation and CO2 forcing, a good discussion on future work should be included beyond a few words. Note that recent studies have reported the non-linear Earth System sensitivity to forcings (e.g., Friedrich et al., 2016, Nature).*

We have added a few lines to Sect. 2.3 (Methodology, Matrix Method) justifying the use of the current climate matrix (containing the 9 HadCM3 snapshots by Dolan et al., 2015), even with its relatively sparse information for warmer-than-present climates, based on the fact that earlier work – Berends et al., 2018 – with a matrix of only 2 states still produced good results. We have also added a few lines to the Discussion section describing the possible benefits of expanding the climate

matrix with additional GCM snapshots, with higher CO2, different ice sheet geometries or different orbital configurations, in future work, which will likely reduce uncertainties.

*4) Finally, reviewer 2 has pretty good comments on discussing model limitation (especially the resolution and*
5 *parameter sensitivity, whether bias correction is used etc), and many detailed comments on improving the clarity of the manuscript. Please do follow them when revising your manuscript.*

We agree with the editor and with anonymous referee #2 that the manuscript was overly brief in describing the ice-sheet model, again relying too much on references to earlier work. We have added a few lines to Sect. 2.2 (Methodology, ice-sheet model)
10 explaining the way sub-shelf melt, ice velocities and the resulting grounding line migration are simulated by the model.

In addition, we took all minor comments into account and we thereby hope that the manuscript improved considerably in clarity.

[revised manuscript text omitted]